# Repeat vaccination reduces antibody affinity maturation across different influenza vaccine platforms in humans

Surender Khurana [1], Megan Hahn[1], Elizabeth M. Coyle[1], Lisa R. King[1], Tsai-Lien Lin[2], John Treanor[3], Andrea Sant[3] & Hana Golding[1]

Several vaccines are approved in the United States for seasonal influenza vaccination every year. Here we compare the impact of repeat influenza vaccination on hemagglutination inhibition (HI) titers, antibody binding and affinity maturation to individual hemagglutinin (HA) domains, HA1 and HA2, across vaccine platforms. Fold change in HI and antibody binding to HA1 trends higher for H1N1pdm09 and H3N2 but not against B strains in groups vaccinated with FluBlok compared with FluCelvax and Fluzone. Antibody-affinity maturation occurs against HA1 domain of H1N1pdm09, H3N2 and B following vaccination with all vaccine platforms, but not against H1N1pdm09-HA2. Importantly, prior year vaccination of subjects receiving repeat vaccinations demonstrated reduced antibody-affinity maturation to HA1 of all three influenza virus strains irrespective of the vaccine platform. This study identifies an important impact of repeat vaccination on antibody-affinity maturation following vaccination, which may contribute to lower vaccine effectiveness of seasonal influenza vaccines in humans

[1] Division of Viral Products, Center for Biologics Evaluation and Research (CBER), FDA, Silver Spring, MD 20993, USA. [2] Division of Biostatistics, Center for Biologics Evaluation and Research (CBER), FDA, Silver Spring, MD 20993, USA. [3] University of Rochester Medical Center, Rochester, NY 14642, USA. Correspondence and requests for materials should be addressed to S.K. (email: Surender.Khurana@fda.hhs.gov)

The annual seasonal influenza vaccines contain one strain each of type A influenza H1N1 and H3N2 viruses, and one or two type B influenza viruses are recommended by the World Health Organization (WHO) and regional health authorities. Several vaccines produced using different platforms have been licensed in the United States. Fluzone (Sanofi Pasteur) represents the split virus egg-based vaccine. FluCelvax vaccine (Seqirus) is produced in Madin Darby canine kidney (MDCK) cells[1,2], and FluBlok (Protein Sciences) is a recombinant hemagglutinin (HA) protein-based vaccine produced in Sf9 insect cells[1-4]. The three vaccine platforms differ in manufacturing processes, viral inactivation (Fluzone and FluCelvax), downstream purification of the HA, and the composition of the final products. Importantly, Fluzone and FluCelvax contain 15 µg of HA (A/H1N1, A/H3N2, B), while FluBlok contains 45 µg of recombinant HA for each strain. Production of influenza vaccines in the different cell substrates also result in different glycosylation patterns[5-7]. Furthermore, cell-substrate adaptation of the vaccine strains to achieve high growth and high HA yield by repeat passages, often selects for amino acid mutations in the HA that can impact the antigenicity or immunogenicity of the final vaccine products[8-10]. In addition, while FluBlok contains pure HAs, the Fluzone and FluCelvax vaccines contain additional viral proteins (neuraminidase, NP, M1) as well as different host cell-derived proteins, which are not measured routinely and are not included in the release specifications. These viral and cellular proteins are likely to contribute to the immune response by eliciting antibodies, CD4, and CD8 T cell-specific responses. Therefore, the quality of the immune response generated following human vaccination by these three seasonal influenza vaccine platforms may differ.

To better understand the impact of different vaccine platforms on the immune response a comprehensive comparative vaccine study was initiated at the University of Rochester, New York Center for Excellence in Influenza Research (NYICE), designed to probe the immune responses following vaccination with egg-based (Fluzone), mammalian cell-based (FluCelvax), and insect cell-based (FluBlok) vaccines during two consecutive influenza vaccine seasons (2015–2016 and 2016–2017). In addition to measuring hemagglutination inhibition (HI), samples from all subjects (Day 0 and Day 28 post-vaccination) are being evaluated for T cell and B cell immune responses.

Here we report the HI titers and in-depth analyses of the real-time antibody-binding kinetics to individual HA domains, HA1 and HA2, using surface plasmon resonance (SPR) following vaccination with products manufactured using the three vaccine platforms and monitoring the impact of repeat/prior vaccination on the quality of the humoral immune response. In addition to measurements of total antibodies against the different HA domains, we measured the affinity of the polyclonal serum antibodies before and after vaccination as previously described for pandemic influenza vaccines[11,12]. Technically, since antibodies are bivalent, the proper term for their binding to multivalent antigens like viruses is avidity, but here we use the term affinity throughout since we measured primarily monovalent interactions.

## Results

**Study population.** A total of 31 subjects in year 1 and 70 subjects in year 2 provided study samples. Of them, 16 subjects participated in both years. Despite randomization, some imbalance among the vaccine groups exists with regard to their demographic and baseline characteristics (Supplementary Table 1). The FluBlok group had more females. The Fluzone group had a higher percentage of subjects vaccinated in the previous year, and thus had a higher pre-vaccination baseline HI titer against H1N1. The percentages of subjects who did not have seroprotective titers (≤40) against H1N1pdm09 before vaccination are similar for the FluBlok and FluCelvax groups, but in the Fluzone vaccine groups all subjects had seroprotective HI titers against H1N1pdm09 (≥40) in both years (Table 1). The percentages of subjects with pre-existing seroprotective titers against H3N2 and B strains are more evenly distributed among the three vaccine groups, especially in year 2 (Table 1).

**Antibody responses with three influenza vaccine platforms.** The goal of the current study was both to better understand the quality of the immune responses to seasonal influenza vaccines generated by three different vaccine platforms licensed based on

---

**Table 1 Distribution of subjects and frequency of responders in this study**

| Influenza subtype | #Seroneg @ Day 0[a], n (%) | Responders (≥4-fold increase)[c], n (%) | #Seropos @Day 0[b], n (%) | Responders (≥4-fold increase)[c], n (%) | #Seroneg @Day 0[a], n (%) | Responders (≥4-fold increase)[c], n (%) | #Seropos @Day 0[b], n (%) | Responders (≥4-fold increase)[c], n (%) |
|---|---|---|---|---|---|---|---|---|
| **FluBlok** | | | | | | | | |
| | **Year 1 (n = 10)** | | | | **Year 2 (n = 22)** | | | |
| H1 | 3 (30%) | 3 (100%) | 7 (70%) | 6 (86%) | 4 (18%) | 2 (50%) | 18 (82%) | 13 (72%) |
| H3 | 3 (30%) | 3 (100%) | 7 (70%) | 6 (86%) | 1 (5%) | 1 (100%) | 21 (96%) | 10 (48%) |
| B | 3 (30%) | 1 (33%) | 7 (70%) | 4 (57%) | 7 (32%) | 1 (14.3%) | 15 (68%) | 3 (20%) |
| **FluCelvax** | | | | | | | | |
| | **Year 1 (n = 12)** | | | | **Year 2 (n = 26)** | | | |
| H1 | 4 (33%) | 4 (100%) | 8 (67%) | 4 (50%) | 2 (8%) | 2 (100%) | 24 (92%) | 13 (54%) |
| H3 | 3 (25%) | 2 (67%) | 9 (75%) | 6 (67%) | 2 (8%) | 2 (100%) | 24 (92%) | 15 (63%) |
| B | 3 (25%) | 2 (67%) | 9 (75%) | 2 (22%) | 6 (23%) | 5 (83%) | 20 (77%) | 3 (15%) |
| **Fluzone** | | | | | | | | |
| | **Year 1 (n = 9)** | | | | **Year 2 (n = 22)** | | | |
| H1 | 0 (0%) | 0 (0%) | 9 (100%) | 5 (56%) | 0 (0%) | 0 (0%) | 22 (100%) | 5 (23%) |
| H3 | 6 (67%) | 6 (100%) | 3 (33%) | 3 (100%) | 2 (9%) | 1 (50%) | 20 (91%) | 4 (20%) |
| B | 3 (33%) | 3 (100%) | 6 (67%) | 3 (50%) | 5 (23%) | 1 (20%) | 17 (77%) | 4 (24%) |

[a]Seronegatives are defined as individuals with pre-vaccination (Day 0) HI titers of <40
[b]Seropositives are defined as individuals with pre-vaccination (Day 0) HI titers of >40
[c]Responders are defined as >4-fold increase in post-vaccination (Day 28) titers over pre-vaccination (Day 0) titers

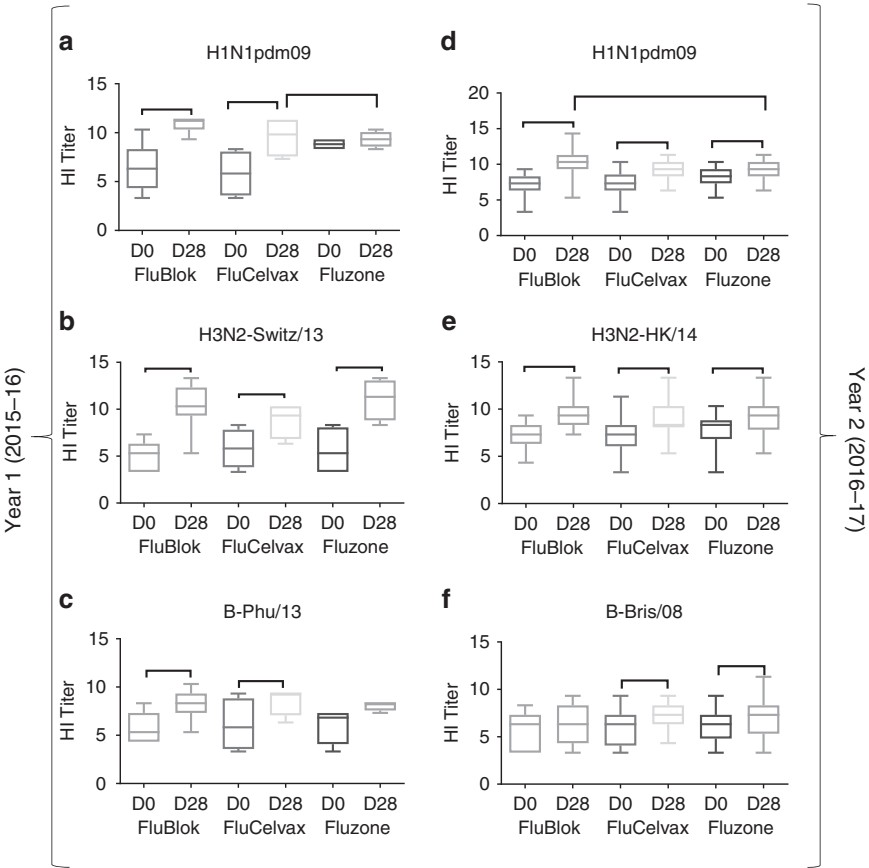

**Fig. 1** HI titers following vaccination with FluBlok, FluCelvax, and Fluzone. **a–f** Average HI titers against H1N1pdm09 (**a**, **d**), H3N2 (**b**, **e**), and influenza B (**c**, **f**) virus strains pre-vaccination (D0) and post-vaccination (D28) are shown for FluBlok, FluCelvax, and Fluzone (excluding repeat vaccinators, see Table 2), in the first (2015–2016) year (**a–c**) and second (2016–2017) year of the study (**d–f**). The box and whisker plot show the median value and the minimum and maximum values. For the Day 28 post-vaccination HI titers and fold change from Day 0, an ANCOVA model was used for comparison between the vaccine groups, adjusted for gender, age, and baseline (Day 0) values (Supplementary Tables 2 and 3). Statistically significant differences between groups and pairs are indicated by horizontal bars. Source data are provided as a Source Data file

clinical efficacy and immunogenicity in the United States, and to determine the impact of repeat/prior vaccination on the quality of these immune responses generated following intramuscular vaccination. Subjects were randomized at the time of enrollment using an internet-based block randomization scheme. Randomization was stratified by self-reported vaccination history in the previous year. In year 2 of the study, participants who had participated in year 1 were re-enrolled and assigned a new subject ID number, but received the same vaccine type that they received in year 1 of the study.

Study participants in year 1 and 2 were randomly divided into three groups that were vaccinated with FluBlok (recombinant hemagglutinin produced in Sf9 insect cells; 45 µg HA/strain/dose)[3,4,13], FluCelvax (cell-based vaccine; 15 µg HA/strain/dose)[1,14,15], and Fluzone (egg-based vaccine; 15 µg HA/strain/dose). HI titers are measured at Day 0 (pre-vaccination) and on Day 28 (Fig. 1). In year 2, the post-vaccination HI GMTs at Day 28 for H1N1 and H3N2 are higher in the FluBlok group, compared to FluCelvax and Fluzone, but only the difference in Day 28 HI titer against H1N1pdm09 between FluBlok and Fluzone is statistically significant (Supplementary Table 2). The results in year 1 are not reliable due to very small sample sizes and a few extreme values. The post-vaccination HI titer fold changes from pre-vaccination baseline (Day 28/Day 0) showed modest increases and are statistically significant, except for the B strain in the FluBlok group in year 2 and the H1N1 and B strains

in the Fluzone group in year 1 (Supplementary Table 3, and Fig. 1a, b and d, e for year 1 and year 2, respectively). In year 2, fold change in HI titer against H1N1pdm09 induced by FluBlok is also significantly higher than FluCelvax and Fluzone (Supplementary Table 3).

**Antibody binding to HA1 and HA2 domains using SPR.** Since the HA1 (globular head) and HA2 (stalk) domains of the influenza virus hemagglutinin undergo differential evolution (i.e., accumulation of mutations under immune pressure)[16,17], it is important to measure total antibody binding as well as antigen–antibody complex dissociation rates as a measure of antibody affinity separately against the two HA domains. We previously described the production of properly folded HA1 and HA2 domains from multiple influenza strains[18–20]. The proper folding of HA0 (produced using insect cells) and bacterially produced recombinant HA1 and HA2 proteins used in this study are confirmed by binding of these proteins to conformation-dependent-neutralizing MAbs specific for either head or stem domain (Supplementary Fig. 1). Furthermore, the functional activity of HA0 and HA1 proteins was confirmed by a hemagglutination assay, which requires the presence of oligomeric HA structures as previously demonstrated[19–22] (Supplementary Fig. 2). Importantly, the HA0, HA1, and HA2 proteins are captured on the chip surface at a low density that ensured monovalent binding of the polyclonal antibodies in the human sera.

Our goal was to conduct in-depth real-time antibody kinetic analyses on the quality of the polyclonal antibody responses. To that end, serially diluted sera at 10-, 50-, and/or 250-fold dilutions are injected at a flow rate of 50 μL/min (300-s contact time) for association, and dissociation was performed over a 600 s interval (at a flow rate of 50 μL/min) (Supplementary Fig. 3). Total antibody binding was determined directly from the serum sample interaction with rHA0, rHA1, and rHA2 protein domains of the influenza virus by SPR as described before[11]. Antibody off-rate constants, which describe the fraction of antigen–antibody complexes that decay per second are determined directly from the serum/plasma sample interaction with rHA0, rHA1, or rHA2 in the dissociation phase only for the sensorgrams with Max RU in the range of 20–150 RU for each serum using BioRad ProteOn SPR machine (Supplementary Fig. 3). Furthermore, to ascertain the antibody kinetics measured under optimized SPR conditions represent primarily the monovalent interactions between the antibody–antigen complex, IgG are purified from the post-vaccination serum and used to prepare Fab molecules. Serial dilution of the IgG and the corresponding Fabs from two serum samples are evaluated for binding to HA1 in the SPR. The antigen–antibody binding off-rates of the IgG and Fab interaction with H1N1pdm09 HA1 from the two serum samples are very similar in spite of the difference in the size (molecular weight) of the bound IgG and Fab molecules (Supplementary Fig. 4).

In the current study, we evaluated responses against all the three HA1 domains from H1N1pdm09, H3N2, and B strains that matched the strains contained in the corresponding seasonal influenza vaccine in each year. In addition, we evaluated binding to the HA0 and HA2 of H1N1pdm09, since this strain was reported to elicit strong anti-stalk antibodies during the 2009 H1N1 pandemic[23,24]. Furthermore, recent study demonstrated accumulation of mutations in both the HA1 head and HA2 stalk of H1N1pdm09-like viruses isolated from infected individuals in the 2015–2016 season[25,26]. We also measured antibody binding to the HA domains on Day 180 following vaccination for year 1 study participants (no such samples available from year 2 study).

Total antibody binding of post-vaccination vs. pre-vaccination sera to intact HA0 and to the HA1 and HA2 domains of H1N1pdm2009 is shown in Fig. 2. In all groups, vaccination induced a moderate to high increase of antibody binding to the H1N1pdm09 HA0, which gained statistical significance only in year 2 for the FluBlok and FluCelvax groups (Supplementary Table 4, Fig. 2a, f). FluBlok generated a significantly higher HA0-binding antibodies compared with Fluzone in year 2 (Fig. 2f, Supplementary Table 4). When comparing binding separately against the HA1 and HA2 antigenic domains, a statistically significant increase in binding to the H1N1pdm09 HA1 was observed following vaccination for the FluBlok group (Supplementary Tables 4 and 5). Furthermore, the maximum antibody binding (Max RU) values following vaccination are significantly higher for FluBlok vaccines compared with FluCelvax and Fluzone vaccinated individuals (Fig. 2b, g for year 1 and year 2, respectively) (Supplementary Table 4). On the other hand, pre-vaccination (Day 0) antibody binding to H1N1pdm09 HA2 domain ise higher than to the HA1 domain of H1N1pdm09, and no significant increase in HA2 binding was found post-vaccination in any of the vaccination groups (Fig. 2c, h vs. b, g). Interestingly, with regard to HA1 from H3N2 strains, vaccination induced increase in total antibody binding for the majority of subjects, irrespective of the vaccine platform. The average post- vs. pre-vaccination binding titers reached statistical significance for the FluBlok groups in both years 1 and 2 and for the Fluzone group only on year 1 (Supplementary Table 4 and Fig. 2d, i, respectively). The antibody-binding levels to the HA1

domains of B strains are variable with many samples demonstrating very high pre-vaccination antibody titers, and minimal or no increase of antibody binding post-vaccination, irrespective of the vaccine platform (Fig. 2e, j). In study year 1, Day 180 post-vaccination samples demonstrated downward trends in antibody binding, but differences are not statistically different from D28 titers for most participants.

These data suggested that the impact of seasonal vaccination is influenced by pre-existing antibodies that bind to both HA1 and HA2 domains. Higher pre-existing antibody titers to H1N1pdm09 HA2 than HA1 are observed, and the main increase in antibody binding following vaccination was focused on the HA1 globular head irrespective of the vaccine platform.

**Antibody affinity maturation following influenza vaccination.** An important attribute of an effective vaccine is the ability to induce germinal center (GC) formation where B cells can undergo affinity maturation/selection resulting in secretion of high-affinity antibodies and generation of long-term plasma cells and memory B cells[27–30]. In SPR, antigen–antibody association kinetics is influenced by both antibody concentration and antibody affinity. However, the dissociation rates of antigen–antibody complexes, under conditions that favor monovalent interaction of each antibody with the HA antigen, primarily reflect the inherent average affinity of the bound polyclonal antibodies[11]. The sensorgrams for representative 10- and 50-fold dilutions of human serum sample are shown in Supplementary fig. 3.

Herein, we measured the antibody affinity in the polyclonal sera by analyzing the antigen–antibody complex dissociation kinetics (off-rates) bound to H1N1pdm09 HA0, HA1, and HA2 as well as to HA1 proteins from the H3 and B strains contained in the seasonal influenza vaccines. Pre-vaccination off-rates varied among subjects and between strains ranging between $10^{-1}\,s^{-1}$ and $10^{-4}\,s^{-1}$ (Fig. 3, D0 open symbols). In the first year, affinity maturation against both HA0 and HA1 of H1N1pdm09 was observed for individuals vaccinated with FluBlok and FluCelvax. However, for the Fluzone vaccine group, the affinity of antibody binding to the HA0 as well as HA1 before vaccination was much higher ($\leq 10^{-3}\,s^{-1}$) in the first year compared with the other two vaccine platforms, and no further affinity maturation was measured in this group following vaccination (Fig. 3a, b, Supplementary Tables 6 and 7). In year 1, increase in binding affinities to HA1 domains of H3 and B strains was observed in most vaccine groups (Fig. 3d, e, respectively; Supplementary Tables 6 and 7). Importantly, by Day 180, antibody dissociation rates against the HA1 domains trended upwards (i.e. faster dissociation of antigen–antibody complexes) indicating loss/decay of the higher affinity antibodies at later time points post-vaccination. This trend was more profound for the anti-H1 HA1-bound antibodies (Fig. 3b).

In the second year, study participants demonstrated heterogeneous binding affinities (off-rates) before vaccination against H1N1pdm09 HA0, and against the HA1 domains of all three strains ranging between $10^{-1}$ to $10^{-4}\,s^{-1}$. Importantly, most individuals with low-affinity pre-vaccination antibodies against the HA1 domains of H1, H3, and B strains ($10^{-1}$–$10^{-2}\,s^{-1}$) demonstrated 10- to 100-fold increase in antibody affinity (i.e., 1–2 log slower dissociation rates) following vaccination, irrespective of the vaccine type received (Supplementary Tables 6, 7, Fig. 3f, g, i, j). FluBlok and FluCelvax induced significant antibody affinity maturation for HA1 of all three strains (Supplementary Table 7, Fig. 3g, i, j).

In contrast to HA1, the binding affinity to the HA2 domain of H1N1pdm09 before vaccination was higher in all three groups (ranging between $10^{-2}\,s^{-1}$ and $10^{-4}\,s^{-1}$). Vaccination did not

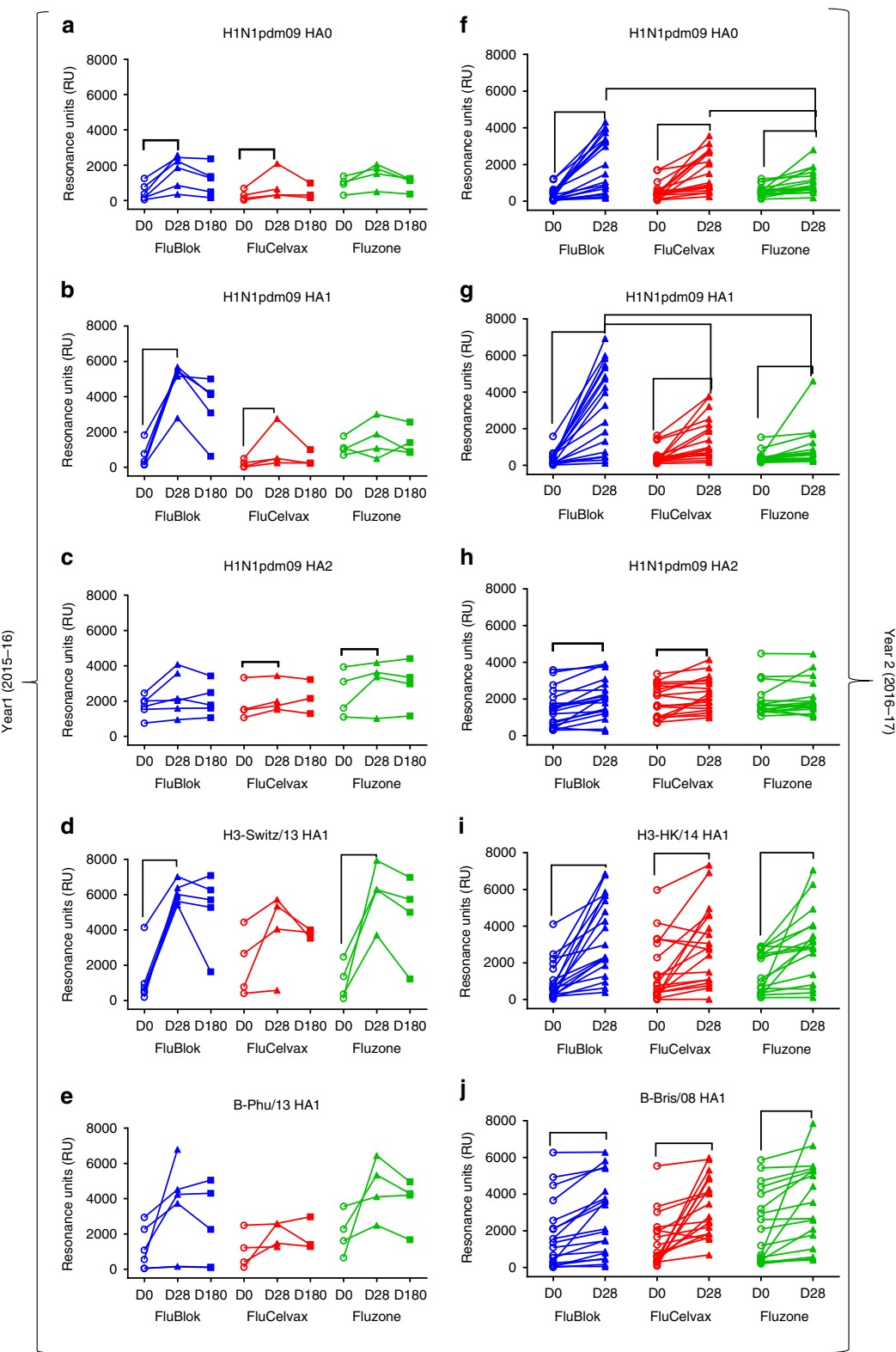

**Fig. 2** Post-vaccination sera binding to native HA0, HA1, and HA2 domains. **a–j** Steady-state equilibrium analysis of human vaccine sera pre-vaccination (D0) and post-vaccination (D28) against properly folded homologous H1N1pdm09 HA0 (**a**, **f**), HA1 (**b**, **g**), and HA2 (**c**, **h**) domains, and H3N2 HA1 (**d**, **i**), and B-HA1 (**e**, **j**) are measured using SPR. Recombinant HA0, HA1, and HA2 domains are immobilized on a sensor chip through the free amine group. Binding of the polyclonal serum antibodies to the immobilized protein is shown as resonance unit (RU) values for individual subjects receiving FluBlok (in blue), FluCelvax (in red), and Fluzone (in green) for each subject (excluding repeat vaccinators) in year 1 (2015–2016; D0, D28, and D180; **a–e**) and year 2 (2016–2017; D0 and D28; **f–j**) of the study. For Day 28 post-vaccination resonance units (RU), and fold change from Day 0, an ANCOVA model was used for comparison between the vaccine groups, adjusted for gender, age, and baseline (Day 0) values (Supplementary Tables 4 and 5). Statistically significant differences between groups and pairs are indicated by horizontal bars. Source data are provided as a Source Data file

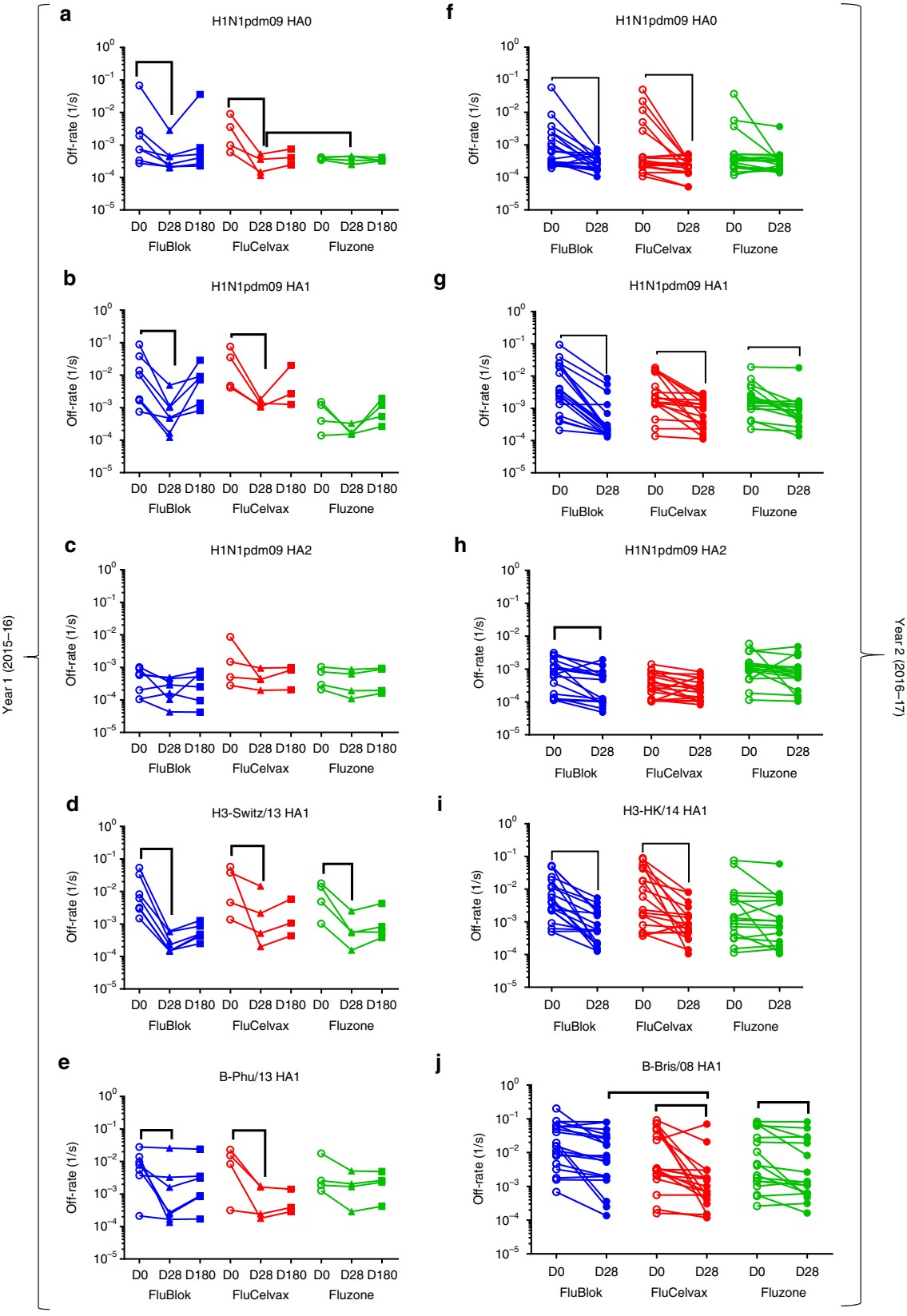

induce discernible affinity maturation against H1N1pdm09 HA2 in the three vaccine groups in the first year. But in the second year, the FluBlok group reached statistical significance for the anti-HA2 affinity change (Fig. 3c, h, respectively; Supplementary Table 7).

In most individuals, there seems to be a ceiling of antibody affinity (approximately $10^{-4}\,s^{-1}$) to HA domains of all vaccine strains following vaccination. These data demonstrated that measuring antibody binding to isolated HA domains allowed detection of selective affinity maturation towards the HA1

**Fig. 3** Kinetics of antibody affinity maturation following human vaccination. **a–j** Sequential SPR analysis of human vaccine sera (pre- and post- vaccination) was performed against properly folded homologous H1N1pdm09 HA0 (**a**, **f**), HA1 (**b**, **g**), and HA2 (**c**, **h**) domains, and H3 HA1 (**d**, **i**) and B-HA1 (**e**, **j**). Samples from 16 individuals (Table 2) that received repeat vaccination in both year 1 and year 2 (Fig. 4) are not included in the Fig. 3 dataset. Ten-fold, 50- and/or 250-fold diluted individual serum from each participant in the vaccine study at pre-vaccination (D0) and at 28 days or 180 days after immunization (D28, D180) are evaluated as shown for FluBlok (in blue), FluCelvax (in red), and Fluzone (in green) for subjects recruited in the first year (2015–2016; **a–e**) and second year (2016–2017; D0 and D28; **f–j**). Serum antibody off-rate constants are determined as described in Methods. Slower dissociation kinetics (off-rate) of antigen–antibody complex means higher antibody affinity. For Day 28 post-vaccination off-rate constants, and fold change from Day 0, an ANCOVA model was used for comparison between the vaccine groups, adjusted for gender, age, and baseline (Day 0) values (Supplementary Tables 6 and 7). Statistically significant differences between groups and pairs are indicated by horizontal bars. Source data are provided as a Source Data file

**Table 2 Demographics and influenza vaccination of subjects in this study**

|  | Season 1 (2015)[a] | | | | Season 2 (2016)[a] | | | | Total[a] |
|---|---|---|---|---|---|---|---|---|---|
|  | Male | Prior | Female | Prior | Male | Prior | Female | Prior |  |
| FluBlok | 4 (1) | 1 | 6 (2) | 0 | 5 (1) | 4 | 17 (2) | 9 | 32 (3) |
| FluCelvax | 6 (4) | 1 | 6 (4) | 0 | 9 (4) | 3 | 17 (4) | 6 | 38 (8) |
| Fluzone | 5 (2) | 1 | 4 (3) | 0 | 11 (2) | 8 | 11 (3) | 5 | 31 (5) |
| Total | 15 (7) | 3 | 16 (9) | 0 | 25 (7) | 15 | 45 (9) | 20 | 101 (16) |

[a]Numbers in parenthesis indicate individuals who are repeatedly vaccinated with same vaccine in both years. Prior are those subjects who received inactivated seasonal influenza vaccine in the previous year of this study

globular head of all vaccine strains (Fig. 3). In addition, there was no clear advantage of any one vaccine platform in the induction of antibody affinity maturation in human adult population following seasonal influenza vaccination. Limited data from Year 1 study also suggested loss of high-affinity circulating antibodies by 6 months post-vaccination for most participants.

**Repeat vaccination reduces antibody affinity maturation**. In the current study, 16 out of 101 participants received repeat vaccination with the same vaccine formulation in both year 1 and year 2: 3 in the FluBlok group, 8 in the FluCelvax group, and 5 in the Fluzone group (Table 2, right column). We analyzed the impact of repeat influenza vaccination in these adults on HI titers, and polyclonal serum antibody affinities against the HA1 globular domain before and after vaccination.

For the anti-H1N1 response, HI titers increased at least twofold between Day 0 and Day 28 during season 1 in all subjects (except one subject in the FluCelvax group). In season 2, the pre-existing anti-H1N1 HI titers are comparatively higher in 50% of participants on Day 0, compared with pre-existing HI titers in season 1. In almost all individuals the second-year vaccination resulted in a weaker immune response (<4 or no fold change between pre- and post-vaccination HI titers in year 2 compared with year 1) across all vaccine platforms (Fig. 4a). A similar trend of dampened HI responses against the H3N2 and B virus strains was observed in year 2 compared with year 1 for most individuals in all vaccine groups (Fig. 4b, c for H3N2 and B strains, respectively).

Since we observed clear trend for serum antibody affinity maturation following vaccination, in both years of the current study, (Fig. 3), we next evaluated the impact of repeat vaccination on affinity maturation for the 16 individuals who received identical vaccine type in both the years across the three vaccine platforms. Similar to the findings with the entire study participants, most of the 16 subjects in the repeat vaccination subset demonstrated antibody affinity maturation during the first year with higher binding affinity (a 50- to 250-fold slower dissociation rates) of serum antibodies bound to the H1N1pdm09 HA1 globular head between post- vs. pre-vaccination samples (Fig. 4d). However, in the second year before vaccination, the affinity of H1N1pdm09 HA1-binding antibodies from most

individuals are lower than after the first-year post-vaccination, and for several of them are as low as the pre-vaccination antibody affinity in year 1. The loss of high-affinity antibodies was already apparent on Day 180 post-vaccination in the first year of study (Fig. 4d, f). Surprisingly, following second-year vaccination, a decrease in antibody affinity maturation against the H1N1 HA1 domain was observed compared with first year, with minimal change in antibody affinity measured for 11/16 of repeat vaccinated individuals (Fig. 4d, h). The same pattern was observed, irrespective of the vaccine platform, with more pronounced drop in affinity fold change observed for the FluBlok and the Fluzone vaccine recipients compared with FluCelvax (Fig. 4h). No or minimal affinity maturation of antibody binding to the H1N1pdm09 HA2 domain was measured following vaccination with any of the vaccine platforms (Fig. 4e, i).

In the case of H3N2, a more complex pattern of antibody affinity maturation emerged, with 12/16 individuals exhibiting minimal antibody affinity maturation to the H3 HA1 domain in year 2 compared with year 1 following repeat vaccination. However, 4/16 subjects (one in the FluBlok group, one in the FluCelvax group, and two in the Fluzone group) showed affinity maturation after year 2 vaccination that was more pronounced than in year 1 (Fig. 4f, j). In the case of anti-B responses, the antibody-binding affinity maturation to the HA1 domain was minimal for most subjects in either year 1 or year 2. Only one subject in the FluCelvax group showed increased antibody affinity for the B-HA1 in year 2 compared with year 1 (Fig. 4g, k).

**Prior vaccination dampens HI response and antibody affinity**. In light of the findings in the 16 repeat vaccinated individuals who received the same vaccine in two consecutive years, we expanded our analysis to determine if prior vaccination in the previous year (irrespective of vaccine platform) had an impact on the HI fold change or antibody affinity maturation in the other 69 study participants in this study (Table 2). All subjects who reported prior-year vaccination received an egg-based inactivated influenza vaccine in the previous year. For this analysis, we combined samples across year 1 and year 2 since no differences are seen between the 2 years (Fig. 5). Repeat vaccinees who received same vaccine in both years (n = 16) are shown in Fig. 4, and participants with unknown prior year vaccination history

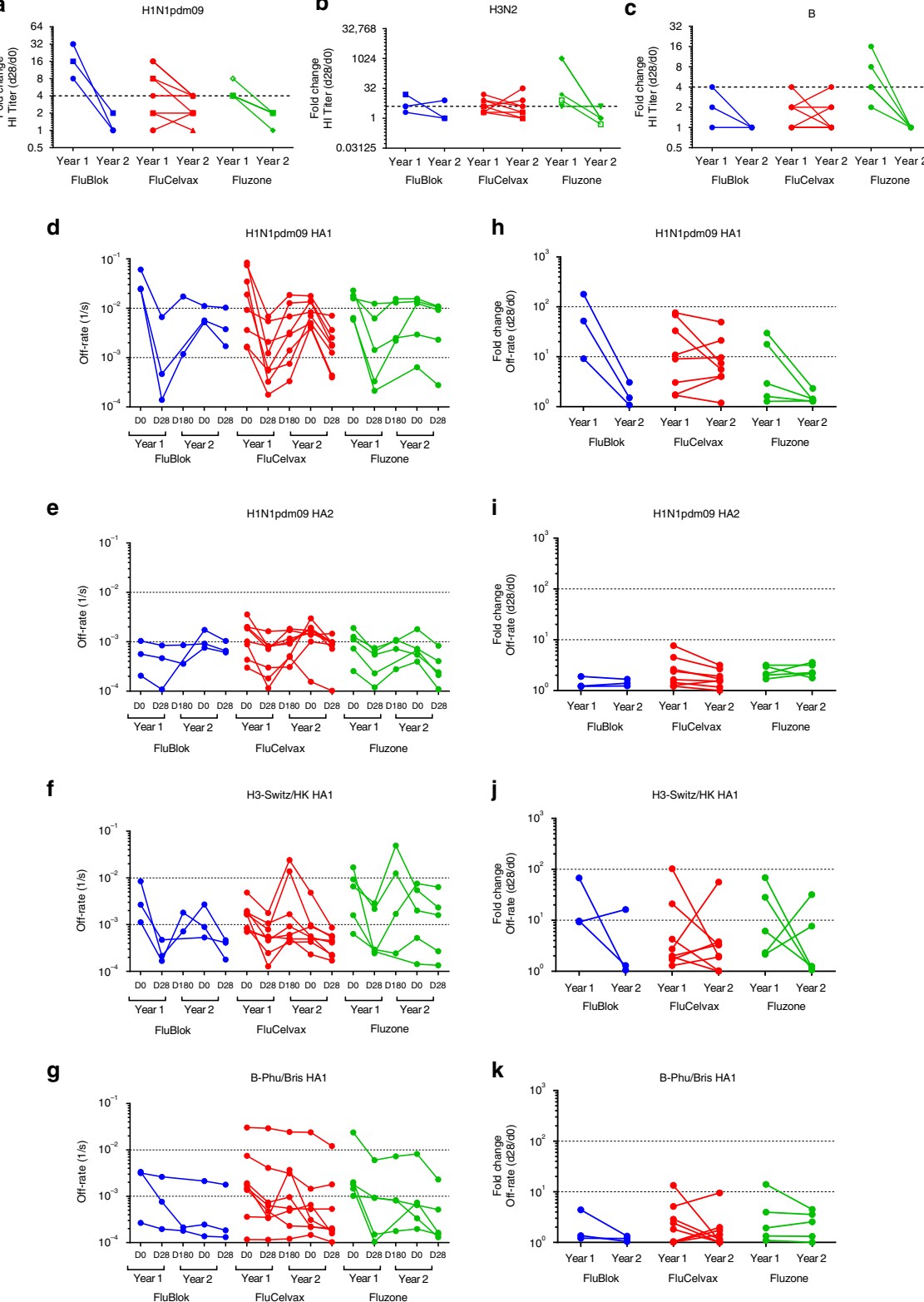

(*n* = 8) are not included in this analysis. Day 0 pre-vaccination HI titers are not statistically different against the three influenza vaccine strains irrespective of the vaccine. Interestingly, only in the FluBlok/FluCelvax groups, the fold increases in HI titers against the H1N1pdm09 strain are higher for individuals with no prior-year vaccination compared with individuals with prior year vaccination (Supplementary Table 8, Fig. 5a). HI responses

against H3N2 are also impacted by prior vaccination more clearly in the FluBlok group (Supplementary Table 8, Fig. 5b). For anti-B response, decreased seroconversion rates are found for individuals reporting prior-year vaccination across all three vaccine platforms, but the decline in fold change reached statistically significant only for the FluBlok and FluCelvax groups (Supplementary Table 8, Fig. 5c).

**Fig. 4** Impact of repeat vaccination on HI titers and antibody affinity maturation. **a–c** Fold change in end-point titers are calculated by dividing post-vaccination (D28) titers by pre-vaccination (D0) titers of respective years for HI against H1N1pdm09 (**a**), H3N2 (**b**), and influenza B (**c**) virus strains as shown for FluBlok (in blue), FluCelvax (in red), and Fluzone (in green) for each of the 16 repeatedly vaccinated individuals. **d–k** Sequential SPR analysis of human vaccine sera (pre- and post- vaccination) was performed against properly folded H1N1pdm09 HA1 (**d**) and HA2 (**e**) domains, and H3 HA1 (**f**) and B-HA1 (**g**). Ten-fold, 50- and/or 250-fold diluted individual serum from 16 repeatedly vaccinated participants at pre-vaccination (D0) and post-vaccination (D28; D180 for year 1 only) are evaluated as shown for FluBlok (in blue), FluCelvax (in red), and Fluzone (in green) for each subject in the first year (2015–2016; S1) and second year (2016–2017; S2). Serum antibody off-rate constants are determined as described in Methods. **h–k** Fold increase in antibody affinity was calculated [1/(off-rate on Day 28/off-rate on Day 0)] for respective years against H1N1pdm09 HA1 (**h**) and HA2 (**i**) domains, H3N2 HA1 (**j**) and B-HA1 (**k**). Source data are provided as a Source Data file

Antibody affinities against H1N1pdm09 HA1 at Day 0 are variable in all vaccine groups, irrespective of prior vaccination status (Fig. 5d). Importantly, antibody affinity maturation to H1 HA1 domain following vaccination was more apparent in those that did not report prior year vaccination, irrespective of vaccine platform, and this trend reached statistical significance for all vaccine groups (Supplementary Table 9, Fig. 5d, h). Prior vaccination status did not impact the affinity of anti-H1 HA2 antibodies at Day 0, and vaccination did not induce discernable affinity maturation of the anti-HA2 antibodies in any of the vaccine platforms (Supplementary Table 9, Fig. 5e, i).

The Day 0 affinities and fold change in antibody affinity against H3 HA1 are also negatively impacted by prior vaccination, and this trend reached statistical significance in the FluBlok as well as FluCelvax vaccinated group (Supplementary Table 9, Fig. 5f, j). With respect to anti-B responses, again a trend towards lower anti-HA1 antibody affinity maturation in individuals with prior-year vaccination was noted for all vaccine groups, but reached statistical significance for the FluBlok and FluCelvax vaccine groups (Supplementary Table 9, Fig. 5g, k).

Next, we analyzed the fold changes in polyclonal serum antibody affinity for all subjects who reported prior-year vaccination (combining subjects in all vaccine groups) and compared them with change in antibody affinity of subjects who did not receive IIV in the prior year (Fig. 6). As can be seen, statistically significant decrease in antibody affinity maturation against the HA1 domains of all vaccine strains (H1N1, H3N2, and B) (but not to HA2 domain of H1N1pdm09; Fig. 6b) was observed (Fig. 6a, c, d).

**HI seroconversion correlates with HA1 antibody affinity.** Finally, correlation was examined between the change in antibody affinity to the isolated HA domains with the functional HI activity of the polyclonal serum antibodies following vaccination of study participants. As shown in Fig. 7, a statistically significant inverse correlation was observed between the HI fold change and the fold change in antibody off-rates of individual study participants in both year 1 and year 2 with the HA1 domains of all vaccine strains and H1N1pdm09 HA0 (Fig. 7a, b, d, e). The fold change in antibody off rates to the HA2 domain of H1N1pdm09 did not correlate with fold change in HI titers (Fig. 7c).

Together, our data revealed a significant negative impact of repeat vaccination (or prior year vaccination) in most study participants on antibody binding, antibody affinity maturation, and HI responses to H1N1, H3N2, and B strains manufactured by three vaccine platforms. Importantly, the lower affinity maturation due to repeat vaccination was focused on the HA1 globular domain, which is the target of most neutralizing/protective antibodies.

## Discussion

In the current study, we compared the immunogenicity of three US-licensed seasonal influenza vaccines manufactured by three different vaccine platforms for intramuscular administration: egg-based inactivated split vaccine (Fluzone), mammalian cell-based inactivated vaccine grown in MDCK cells (FluCelvax), and insect cell-based recombinant HA (FluBlok). The two new platforms (FluCelvax and FluBlok) were previously evaluated in parallel with the egg-based vaccine in young adults and the elderly for safety and immunogenicity (HI seroconversion)[3,31]. However, the current study was designed to compare the immunogenicity of all three vaccines side by side over 2 years, and for in-depth analyses using multiple immunological endpoints. We focused on the analysis of quality of the polyclonal serum antibodies generated following intramuscular vaccination with the different seasonal influenza vaccines using SPR-based real-time kinetics assays to measure total antibody binding as well as antibody affinity maturation against isolated domains (HA1 and HA2 from H1N1pdm09 and HA1 from H3 and B) of the strains contained in the vaccines[11,19].

In several assays, there was a trend towards higher antibody response in subjects receiving the FluBlok vaccine (Figs. 1–3). This finding raised the possibility that the purified recombinant HA vaccine produced in insect cells (lower glycosylation) may have the capacity to boost antibodies with broad cross reactivity. We also used an extended virus neutralization assay to measure the ability of the vaccines manufactured using the three platforms to generate antibodies that cross-neutralize heterosubtypic strains representing group 1 (H5N1 A/Vietnam) and group 2 (H7N9 A/Shanghai) avian viruses. However, no heterosubtypic neutralizing antibody responses were observed irrespective of the vaccine platform.

Therefore, in spite of the fact that the three vaccine products are manufactured in different cell substrates, resulting in different glycosylation patterns and containing different amounts of HA, and host and viral cell proteins, the functional antibody responses showed either no or selective advantage for the FluBlok vaccine, primarily against the H1N1pdm09 strain in the assays described here. However, whether the three vaccine products are processed similarly by antigen-presenting cells and elicit similar (or different) specificities of CD4[+] T helper (Th) and T follicular helper (TFh) cells is currently under investigation. Surprisingly, the three vaccines induced antibody responses that are not significantly different in terms of antibody affinity maturation, even though FluBlok vaccine contains three times more antigen (45 μg HA/dose), which could theoretically lead to induction of more TFh cells, required for GC formation.

The most interesting outcome in the current study was the finding of significant affinity maturation of antibodies targeting the HA1 domain that occurred following vaccination. In contrast, antibody affinity maturation was not evident for antibodies targeting the HA2 domain of H1N1pdm09, which is more conserved than the HA1 globular head domain. Thus, most of the pre-existing HA2-binding antibodies exhibited high affinity (i.e., slow dissociation rates) before vaccination, suggesting that immune response to HA2 is dominated by long-term memory B cells or long-lasting plasma cells. In the ferret model, we observed direct

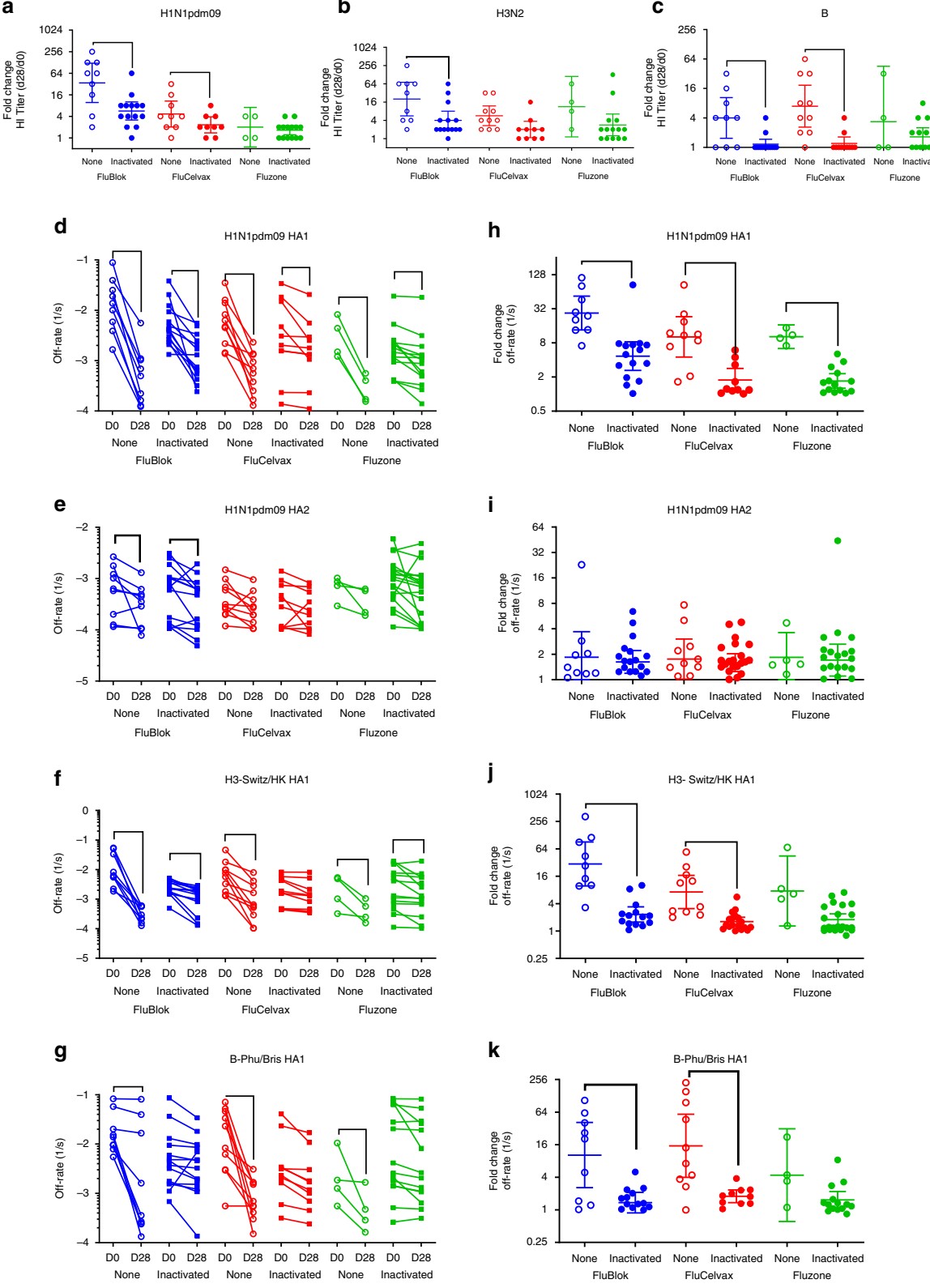

correlation between high-affinity antibodies against the HA1 domain of H7N7 following vaccination and reduction in lung viral loads after challenge with either H7N7 or H7N9 highly pathogenic avian influenza viruses[22]. Furthermore, high titers of low avidity antibodies in infected individuals were associated with severe H1N1pdm09 disease[32].

The negative impact of repeated influenza vaccination on vaccine responsiveness and vaccine effectiveness (protection from infection) received increasing attention in recent years[33–37]. In the current study, only 16 individuals (15%) are vaccinated repeatedly with the same vaccine product in two consecutive years. The significant decline in the HI seroconversion rates in

**Fig. 5** Impact of prior vaccination on HI and antibody affinity maturation. In the current study 38/69 individuals received inactivated seasonal influenza vaccine in the previous year (labeled inactivated), while 23/69 were not vaccinated with influenza vaccine in the previous year (labeled none). **a–c** End-point HI titers against H1N1pdm09 (**a**), H3N2 (**b**), and B (**c**) virus strains at Day 0 (D0; pre-vaccination) and Day 28 (D28; post-vaccination) are shown for FluBlok (in blue), FluCelvax (in red), and Fluzone (in green) for each subject. Fold change in end-point titers are calculated by dividing post-vaccination (Day 28) titers by pre-vaccination (Day 0) for individuals with prior vaccination (Inactivated; closed symbols) vs. no prior year vaccination (None; open symbols), and are shown for each vaccine group. **d–k** Sequential SPR analysis of human vaccine sera (pre- and post- vaccination) was performed against properly folded H1N1pdm09 HA1 (**d**) and HA2 (**e**) domains, H3N2 HA1 (**f**), and B-HA1 (**g**). Ten-fold, 50- and/or 250-fold diluted individual serum at pre-vaccination (D0) and at 28 days after immunization (D28) are evaluated for FluBlok (in blue), FluCelvax (in red), and Fluzone (in green) vaccinated individuals who received inactivated seasonal influenza vaccine in the previous year (Inactivated) and individuals who were not vaccinated with influenza vaccine in the previous year (None). Serum antibody off-rate constants are determined as described in Methods. **h–k** Increase in antibody affinity as measured by fold change in antigen–antibody complex dissociation rates was calculated [1/(off-rate on Day 28/off-rate on Day 0)] against H1N1pdm09 HA1 (**h**) and HA2 (**i**) domains, H3N2 HA1 (**j**) and B-HA1 (**k**) are shown for FluBlok (in blue), FluCelvax (in red), and Fluzone (in green) for each subject. Geometric means with 95% confidence intervals are shown. For pairwise comparison, an ANCOVA model was used for comparison between the vaccine groups, adjusted for gender, age, and baseline (Day 0) values (Supplementary Tables 8 and 9). Statistically significant differences between groups and pairs are indicated by horizontal bars. Source data are provided as a Source Data file

year 2 is not due to antigenic distance, since all vaccine strains were antigenically very similar (or identical) during the 2 years of the study. Importantly, all of the SPR studies were conducted with recombinant proteins produced either in insect cells (H1N1pdm09 HA0) or in *E. coli* (HA1 and HA2 domains) that are identical in sequence to the recommended influenza vaccine strains and did not contain mutations associated with egg adaptation[10,38,39].

More importantly, in spite of the increase in antibody affinity towards HA1 domains following vaccination in year 1, the affinities of antibodies in year 2 before vaccination are lower compared with affinity of antibodies after first-year vaccination. This observation suggested that high-affinity antibodies generated following seasonal influenza vaccination were not long-lasting and could be linked to the reported decline in vaccine efficacy over time following vaccination[36,39,40]. Furthermore, only limited anti-HA1 antibody affinity maturation was observed after the second-year vaccination. This finding was expanded when we compared antibody affinity maturation following vaccination in all individuals who reported prior-year IIV vaccination vs. those that reported no prior vaccination.

These findings suggest that the high-affinity antibodies generated after year 1 vaccination probably are derived from terminally differentiated post-GC plasma cells, in agreement with previous studies[41,42]. Apparently, most of the activated plasma cells are exhausted and undergone apoptosis, with minimal seeding of the bone marrow. Short-term memory B cells that could re-seed the GC along with naïve B cells are also negatively affected by repeat vaccination. Thus, large fraction of high-affinity plasma cells generated after the initial vaccination may be short lived, and the B cell repertoire returns to the baseline before next year vaccination. These antibodies are of lower affinity because they are (or may be) specific for another HA based on previous exposure to influenza antigens (through infection or vaccination). The possibility of other immune mechanisms including immune exhaustion or immune competition cannot be ruled out.

Eidem et al.,[43] previously reported persistence and avidity maturation of antibodies to A(H1N1)pdm09 in healthcare workers (HCW) following repeated annual vaccinations (2009–2011). In that study, HCW were first vaccinated in 2009 with an AS03 adjuvanted H1N1pdm09 vaccine (Pandemrix), followed in 2010 and 2011 by unadjuvanted TIV. While most subjects seroconverted after year 1, 2, and 3 vaccinations, the highest HI and MN GMT were measured after the first-year vaccination. The apparent increase in antibody avidity between 2009 and 2011 (as measured in HA1-ELISA with or without NaSCN treatment) probably suggest that primary vaccination with the adjuvanted vaccine elicited long-term memory B cells

that undergo further maturation and differentiation following year 2 and year 3 vaccinations. These findings are in agreement with our previous studies with oil-in-water adjuvanted pandemic influenza vaccines that demonstrated superior seroconversion titers, expanded epitope repertoires (more antibodies targeting protective epitopes in the HA1 domain), and significantly higher affinity maturation of antibodies in adjuvanted compared with unadjuvanted vaccine groups[11,12,44]. In a long-term prime-boost study, individuals primed with MF59-adjuvanted H5N3 (clade 0) vaccine elicited rapid high titers and high-affinity antibodies when boosted 6 years later with heterologous H5N1 (clade 1) vaccine, in contrast to individuals who were primed with unadjuvanted vaccine[45].

We acknowledge the limitation of the current study that was conducted as an observational descriptive study. Future prospective randomized studies with individuals with known vaccination history will be conducted to further explore the contribution of antigen platform, and antigen dose to antibody affinity maturation. It will be of interest to compare the rates of somatic hypermutations (SHM) in HA1-specific plasma cells isolated early vs. late time points after yearly vaccination. Such approach recently revealed that influenza virus vaccination in the elderly results in less de novo somatic hypermutations in the immunoglobulin variable genes compared with younger adult[46]. It is important to determine if the recruitment of new B cells into GCs is linked to the frequency of TFh cells and how it is influenced by repeated influenza vaccination and the antigenic match between first year and second year strains[42,47]. We hypothesize that the inability to undergo rapid antibody affinity maturation may indirectly reduce the efficiency of blocking virus receptor binding/internalization and virus clearance, which could contribute to the reported lower protection effectiveness-associated repeated vaccination[33–37,39].

This is the first report that compared three US-licensed inactivated influenza vaccines (manufactured by different platforms) side-by-side and demonstrates that antibody affinity maturation was significantly reduced in individuals reporting prior-year seasonal inactivated influenza vaccination or 2-year consecutive vaccination with the same influenza vaccine, and could inform development of better and more effective influenza vaccines. Therefore, it's critical to measure antibody affinity in studies with current and new vaccines strategies, as well as antibody based therapeutics.

## Methods
**Clinical study.** Subjects in this Phase 4 clinical study (ClinicalTrials.gov Identifier: NCT03068949) were healthy adults between 18 and 49 years of age (inclusive). Good health was determined by medical history and targeted physical examination

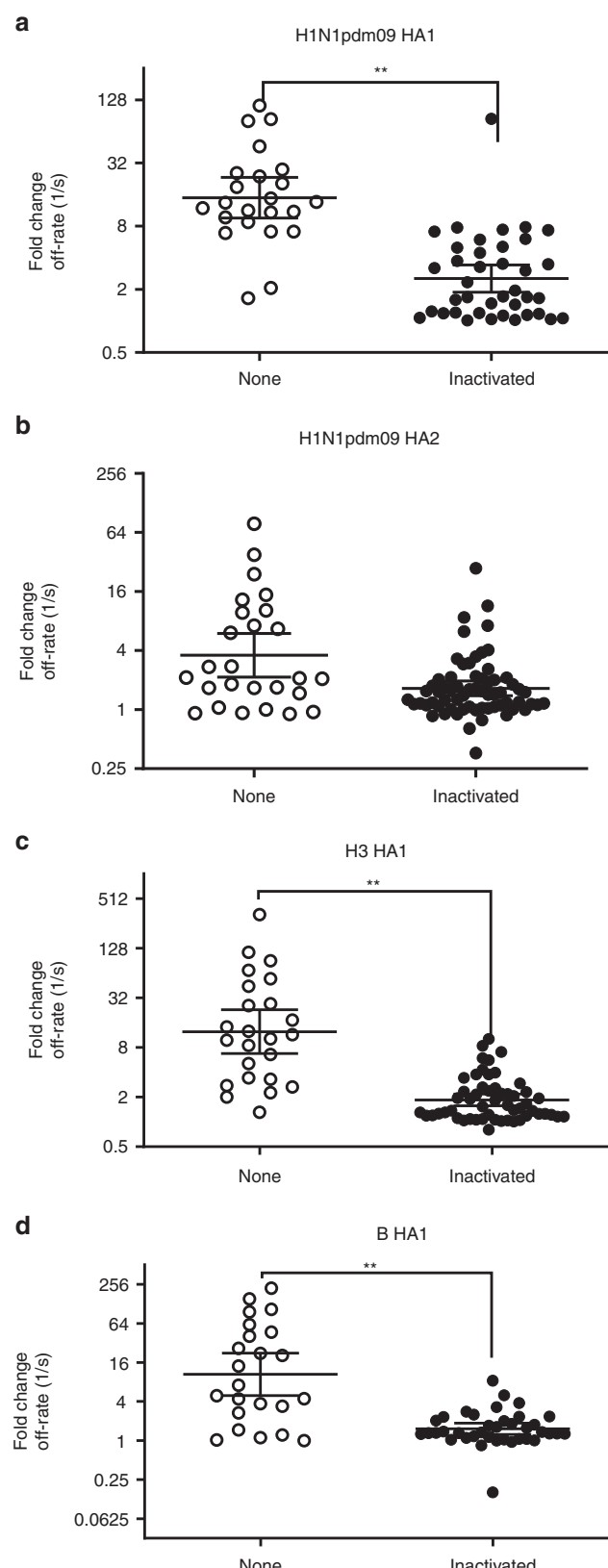

**Fig. 6** Prior year vaccination negatively impacts affinity maturation of antibodies. **a–d** Sequential SPR analysis of human influenza vaccine sera (pre- and post-vaccination) was performed against H1N1pdm09 HA1 (**a**) and HA2 (**b**) domains, H3N2 HA1 (**c**), and B-HA1 (**d**). Ten-fold, 50- and/or 250-fold diluted individual serum at pre-vaccination and following vaccination with FluBlok, FluCelvax, or Fluzone from individuals who received inactivated seasonal influenza vaccine in prior year (Inactivated) and individuals who were not vaccinated with influenza vaccine in the previous year (None). Serum antibody off-rate constants were determined as described in Methods. Increase in serum antibody affinity as measured by fold change in antigen–antibody complex dissociation rates was calculated as described in Fig. 5 [1/(off-rate on Day 28/off-rate on Day 0)] against H1N1pdm09 HA1 (**a**) and HA2 (**b**) domains, H3N2 HA1 (**c**) and B-HA1 (**d**) for each subject. Geometric means with 95% confidence intervals are shown. The pairwise comparison was analyzed using the Mann–Whitney non-parametric method between two groups. The significance between groups that are statistically significant with a *p* value of <0.005 (**) are shown. Source data are provided as a Source Data file

safety or assessment of reactogenicity and immunogenicity and did not indicate a worsening of medical diagnosis or condition. Subjects had normal vital signs at the time of enrollment.

Subjects are excluded if they reported hypersensitivity to components of the study vaccine or other components of the study vaccine or latex allergy, history of severe reactions following previous immunization with licensed or unlicensed influenza virus vaccines, history of Guillain-Barre syndrome within 6 weeks of receipt of a previous influenza vaccine, if they are pregnant or breastfeeding or intended to become pregnant during the study period, if they are immunosuppressed as a result of an underlying illness or treatment with immunosuppressive or cytotoxic drugs, or use of anticancer chemotherapy or radiation therapy within the preceding 36 months or had received immunoglobulin or another blood product within the 3 months prior to enrollment in this study, if they had active neoplastic disease defined as having received treatment within the past 5 years. if they had long-term (greater than 2 weeks) use of oral or parenteral steroids, or high-dose inhaled steroids, or if they had received an inactivated vaccine within the 2 weeks or a live vaccine within the 4 weeks prior to enrollment in this study or plans to receive another vaccine within the next 28 days.

In both year 1 and year 2 of the study, healthy adults aged 18–49 years were randomized to receive one of three US-licensed influenza vaccines: FluBlok, a pure recombinant hemagglutinin influenza vaccine (rHA, Protein Sciences Corp), or Fluzone, a subvirion influenza vaccine made in eggs (Sanofi) or FluCelvax, a MDCK cell culture (Sequirus)-derived vaccine (Table 1, Supplementary Table 1 and Source data file). This study was sponsored by National Institute of Allergy and Infectious Diseases (NIAID) under DMID Protocol Number: 15-0055. The FluBlok vaccine contains 45 µg of each HA antigen for the virus strains contained in the vaccine, while Fluzone and FluCelvax contain 15 µg of each HA antigen for the virus strains contained in the vaccine. However, the potency assays used for FluBlok and Fluzone are different, and the value of 45 or 15 µg, respectively, may not be directly comparable to each other.

In the 2015–2016 season study participants received the trivalent formulation of vaccine containing A/California/7/2009 (H1N1pdm09)-like virus, A/Switzerland/9715293/2013 (H3N2)-like virus, and B/Phuket/3073/2013-like virus. In the 2016–2017 season, the participants received the quadrivalent formulation of Fluzone and FluCelvax vaccine including A/California/7/2009 (H1N1pdm09)-like virus, A/Hong Kong/4801/2014 (H3N2)-like virus, B/Brisbane/60/2008-like virus and B/Phuket/3073/2013-like virus, while the FluBlok vaccine was a trivalent formulation that included A/California/7/2009 (H1N1pdm09)-like virus, A/Hong Kong/4801/2014 (H3N2)-like virus, and B/Brisbane/60/2008-like virus.

In this study, cohort of otherwise healthy adults, many of whom are employees of the University of Rochester, were asked if they had received influenza vaccine in the previous year, and randomization was stratified based on vaccination history. We did not attempt to verify personal vaccine history by review of medical records. In addition, subjects who had participated in the previous year could participate in the second year of the study and received the same vaccine they had received in the first year.

Subjects were randomized at the time of enrollment using an internet-based block randomization scheme. Subjects with self-reported prior vaccination history were randomized separately from those who reported that they did not receive vaccine in the previous year. In year 2 of the study, participants who had participated in year 1 were re-enrolled and assigned a new subject number, but received the same vaccine type that they received in year 1 of the study. The study was late getting off the ground in year 1 and was more of a pilot run for a bigger recruitment effort in year 2. See eligibility criteria above.

to evaluate acute or currently ongoing chronic medical diagnoses or conditions, defined as those that have been present for at least 90 days, which would affect the assessment of the safety of subjects or the immunogenicity of study vaccinations. Subjects could be taking medications if in the opinion of the site principal investigator or appropriate sub-investigator they posed no additional risk to subject

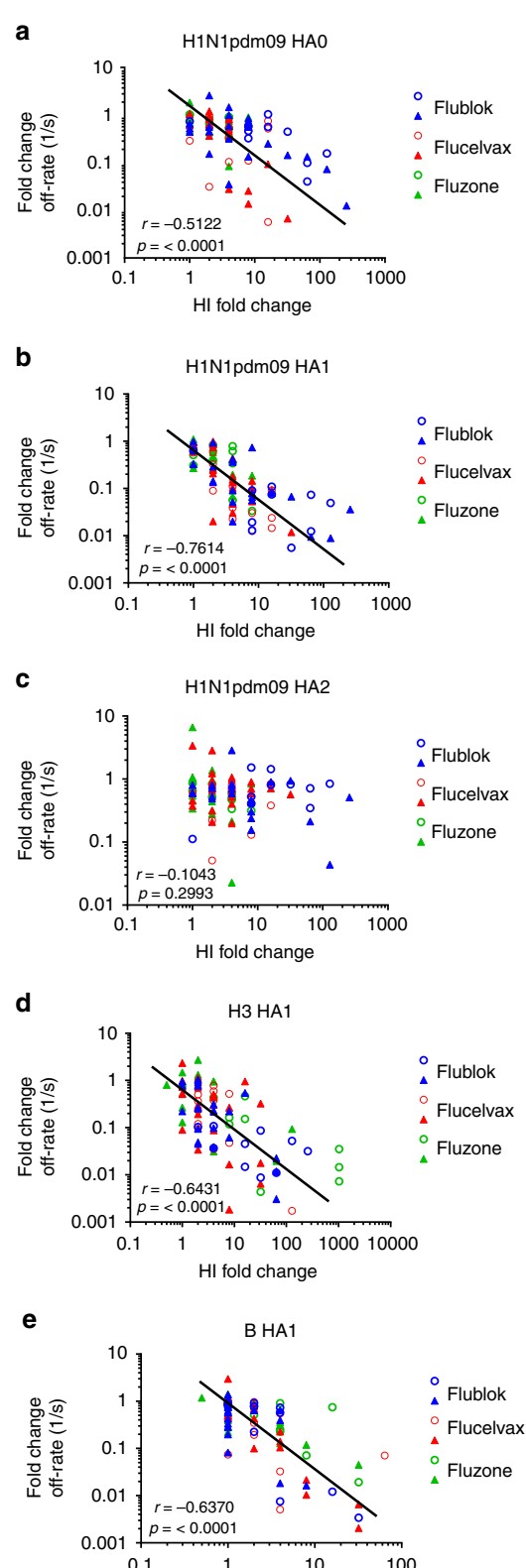

**Fig. 7** Correlations between fold change of HI and anti-HA1 antibody off-rates. **a**–**e** HI fold changes of all study participants are correlated with their change of antibody-binding off-rates against (**a**) H1N1pdm09 HA0; (**b**) H1N1pdm09 HA1; (**c**) H1N1pdm09 HA2; (**d**) H3 HA1; (**e**) B-HA1. Spearman correlations, assuming non-parametric correlation, are reported for the calculation of correlations between off-rate fold changes and HI fold changes across all vaccine groups in year 1 and year 2. The correlation coefficient and $p$ values are reported for each set of all groups Source data are provided as a Source Data file

The study samples were coded, and all the antibody assays were performed blindly. After the data was generated, samples were unblinded to perform the data analysis in this study. The protocols were evaluated by CBER/NIH Research Involving Human Subjects Committee and conducted under RIHSC exemption #03-118B.

**Ethics statement**. The study at CBER, FDA was conducted with de-identified samples under Research Involving Human Subjects (RIHSC) exemption #03-118B, and all assays performed fell within the permissible usages in the original consent.

**Serum HI**. HI assays were performed in microtiter format using turkey RBCs and egg-grown, A/California/07/2009 (H1N1pdm09), A/Switzerland/9715293/13 (H3N2), B/Phuket/3073/13 (Yamagata Lineage), and B/Brisbane/60/08 (Victoria Lineage) viruses[48]. The HI titer of serum antibodies was defined as the highest dilution resulting in complete inhibition of hemagglutination. Sera were treated with receptor-destroying enzyme and heat inactivated prior to testing at an initial starting dilution of fourfold. Sera with no detectable HI titer are assigned a titer of 2 for data analysis purposes.

**Generation of rHA1 and rHA2 recombinant proteins**. The DNA gene segments corresponding to the HA1 and HA2 proteins of A/California/07/2009 (H1N1pdm09) virus, A/Switzerland/9715293/13 (H3N2), A/Hong Kong/4801/2014 (H3N2), B/Phuket/3073/13 (Yamagata Lineage), and B/Brisbane/60/08 (Victoria Lineage) are cloned as NotI–PacI inserts into a T7 promoter-based pSK expression vector in which the desired polypeptide can be expressed as a fusion protein with $His_6$ tag at the C-terminus. The HA1 and HA2 proteins are expressed and purified as described before[18–20]. Briefly, E. coli Rosetta Gami cells (Novagen) are used for expression of HA1 and HA2 proteins. Following expression, inclusion bodies are isolated by cell lysis and multiple washing steps, denatured, refolding in redox-folding buffer, and dialyzed. The dialysate was filtered through a 0.45 μm filter and was subjected to purification by HisTrap Fast flow chromatography. The purified proteins are characterized for the presence of oligomers by gel-filtration chromatography and by functional binding to RBC in a hemagglutination assay. The glycosylated HA0 of H1N1pdm09 produced in Sf9 insect cells was purchased from Protein Sciences Inc.[49].

The quality of the refolded purified HA1 proteins was confirmed by SPR using conformation-sensitive neutralizing monoclonal antibodies against HA of H1N1 (MAb M19B5 against HA head) (Supp. Fig. 1a), H3N2 (MAb 3G6, Supp. Fig. 1C), and B (MAb 5A1, Supp. Fig. 1D) influenza strains. In addition the stem targeting bnMAb CR6261 (ref. [50]) bound to the recombinant H1N1pdm09 HA0 and HA2 (Supp. Fig. 1B). Neutralizing mouse monoclonal antibodies M19B5 (anti-H1N1), 3G6 (anti-H3N2), and 5A1 (anti-B) were a kind gift from Dr. Jerry Weir (Division of Viral Products, CBER, FDA).

To confirm that the rHA1 proteins also formed oligomers (similar to the native spike hemagglutinins on virions) hemagglutination assay was performed with human red blood cells (RBC). The HA0 from H1N1pdm09 and all the purified rHA1 proteins from different strains agglutinated RBC to various concentrations, while the HA2 domains did not agglutinate RBC (Supplementary Fig. 2). We previously demonstrated that several recombinant HA1 domains produced using bacterial system resembled native viral HA in EM, formed functional trimers/oligomers that were fully immunogenic, generated high-affinity antibodies, and protected ferrets from influenza challenge with pandemic strains[18–21]. The HA1 domains could also adsorb the majority of neutralizing antibodies from post-vaccination polyclonal antibodies in human plasma[11].

**Purification of IgG from serum and Fab molecules**. IgG was purified from serum using Protein A chromatography per the manufacturer's instructions (Pierce/Thermofisher). Purified IgG was digested with Papain and the cleaved Fc was removed using Nab Protein A Plus Spin column kit (Thermofisher) and Fab fraction was collected as the flow-through fraction.

**Binding kinetics of serum antibodies to HA1 and HA2 by SPR**. Steady-state equilibrium binding of post-vaccination individual sera was monitored at 25 °C using a ProteOn surface plasmon resonance biosensor (BioRad)[11,12,51]. The rHA1

For SPR, based on multiple prior human studies, and assuming similar variability in change in antibody binding and antibody affinity over time, it was calculated that we will have adequate power to detect meaningful differences in antibody binding/affinity with 20–30 participants in each of the described categories (81% for $n = 20$ and 93% for $n = 30$). In the second year (2016–2017), the number of participants for each vaccine modality was 22–26 per vaccine arm (Table 2).

and/or HA2 proteins from the corresponding year vaccine strains were coupled to a GLC sensor chip with amine coupling with 500 resonance units (RU) in the test flow cells. Samples of freshly prepared sera at 10-, 50- and/or 250-fold dilutions were injected at a flow rate of 50 µL/min (300 s contact time) for association, and dissociation was performed over a 600 s interval (at a flow rate of 50 µL/min) (Supp. Fig. 3). Responses from the protein surface were corrected for the response from a mock surface and for responses from a separate, buffer only injection. MAb 2D7 (anti-CCR5) was used as a negative control in these experiments. Total antibody binding was determined directly from the serum sample interaction with rHA1 and rHA2 protein domains of the influenza virus by SPR using the BioRad ProteOn manager software as described before. Antibody off-rate constants, which describe the fraction of antigen–antibody complexes that decay per second, are determined directly from the serum/plasma sample interaction with rHA0, rHA1, or rHA2 using SPR in the dissociation phase only for the sensorgrams with Max RU in the range of 20–150 RU and calculated using the BioRad ProteOn manager software for the heterogeneous sample model as described before[11]. Off-rate constants are determined from two independent SPR runs.

To confirm that the intact polyclonal IgG interacts with HA via monomeric interaction under the defined SPR conditions, binding kinetics of purified IgG from post-vaccination sera and Fab fragments were compared. To that end, serial dilutions for each of purified IgG and purified Fab fractions of serum sample were analyzed for binding to HA1 proteins under optimized conditions in SPR as described above (Supplementary Fig. 4).

**Statistical analyses.** For Day 28 post-vaccination HI titers, resonance units (RU), and off-rate constants, an ANCOVA model was used for comparison between the vaccine groups, adjusted for gender, age, and baseline (Day 0) values. For fold changes (Day 28/Day 0) in HI titer, RU, and off-rate, an ANCOVA model which includes gender and age as covariates was used for comparison between the groups. The estimated mean fold changes for each vaccine group were also tested for significance. These analyses were performed separately for each year on subjects without repeat vaccinations in both years. The year 1 results were not reliable due to very small sample sizes after removing subjects with repeat vaccination. Analyses were performed on the logarithmic scale, as normality is generally achieved by logarithmic transformation for immunogenicity data. Least-square means for each vaccine group and estimated between-group differences, along with their corresponding 95% confidence intervals, are back-transformed to obtain the geometric means for each vaccine group and the geometric mean ratios between groups. Bonferroni adjustment was used to control type I error for multiple comparisons. The effect of prior vaccination on immune response was evaluated by comparing the fold changes in HI titer and off-rate between subjects received inactivated influenza vaccine in the previous year vs. none for each vaccine arm, and with year 1 and year 2 combined. Spearman correlations are reported for the relationship between off-rate fold changes and HI titer fold changes across all vaccine groups in year 1 and year 2. Statistical analyses were performed using SAS 9.4 and GraphPad software.

**Reporting Summary.** Further information on research design is available in the Nature Research Reporting Summary linked to this article.

## Data Availability
The datasets generated during and/or analyzed during the current study are available from the corresponding author on reasonable request. The Source data with all the data used in figures and tables are provided as a Source Data file in this manuscript.

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

## Acknowledgements

We thank David Topham, Keith Peden, and Zhiping Ye for their insightful review of the manuscript. Neutralizing mouse monoclonal antibodies M19B5 (anti-H1N1), 3G6 (anti-H3N2), and 5A1 (anti-B) were a kind gift from Dr. Jerry Weir (Division of Viral Products, CBER, FDA). This project has been funded in whole or in part with Federal funds from the National Institute of Allergy and Infectious Diseases, National Institutes of Health, Department of Health and Human Services, under CEIRS Contract No. HHSN272201400005C.

## Author contributions

Designed research: S.K., Performed research: M.H., E.C., L.R.K., J.T., and S.K., Statistical analysis: T.-L.L., Conducted phase 4 study and contributed clinical samples: J.T. and A.S., Contributed to writing: S.K., J.T., A.S., and H.G.

## Additional information

**Competing interests:** The authors declare no competing interests.

