## [Peer Review File · Nature Communications]

Reviewers' Comments:

Reviewer #1:

Remarks to the Author:

The manuscript submitted by Khurana et al. is well written, and the logic is easy to follow. The study describes findings relating to affinity maturation in the context of serial vaccination using three seasonal influenza vaccines, based on egg produced antigens, cell line produced antigens, and a vaccine used on a recombinant product. No major difference was observed in terms of overall HI or binding titers (while the FluBlok trended a bit higher, this vaccine also has a higher dose of antigen). The authors conclude that affinity maturation could be observed for all three vaccines (and each of the three components) against the HA1, but not against HA2. Finally the authors find reduced affinity maturation in the second season studied, focusing on the repeat vaccinees. The authors conclude that they have identified an important impact of repeat vaccination on antibody-affinity maturation following vaccination, which may contribute to lower vaccine effectiveness of seasonal influenza vaccines.

While the study is well executed and described, I have several conceptual concerns about the design and interpretation of these experiments.

1. The use of such an early post-vaccination timepoint (day 28) samples to address this issue has some inherent problems. Multiple groups have studied plasma blast responses after influenza vaccination, and it stands to reason that a lot of the serum antibody observed at day 28 derives from these cells. As far as I am aware, it remains unclear if these cells have undergone a germinal center reaction, or represent short-lived, acutely induced cells. In my opinion, using a later time point for these studies would be more informative for the conclusions made in this manuscript.
2. The lack of affinity maturation in the repeat vaccinees is only based on 16 donors, spread across the three different vaccine groups. Given the inherent variability of the data this might be a little on the low end for the interpretations made.
3. Without additional repertoire analyses of vaccine induced cells, it is not clear if the observed differences derives from a lack of affinity maturation, or recruitment of a different set of B cells into the response the second year. And again, if the findings observed derive from long or short lived plasma cells. This concern is amplified by the findings shown in figure 4D, where the off rate is reduced after vaccination (affinity maturation) in season 1, but then at day 0 for season 2, seems to have come back up to a similar value as for baseline of year 1.
4. Finally, the use of the bacterially produced antigen constructs is a concern. While the authors do show binding of a handful of monoclonal antibodies to key epitopes of the recombinant proteins, it is not clear that they truly represent the entire repertoire of epitopes on the influenza virion, possibly incorporating additional uncertainty.

Reviewer #2:

Remarks to the Author:

In this manuscript, "Comparison of antibody responses following different influenza vaccine platforms in humans: Repeat vaccination reduces antibody affinity maturation," Khurana et. al. interrogate a very unique set of human sera from individuals immunized over two consecutive years with three different licensed seasonal influenza vaccines, each one produced via a different platform. Egg-based and mammalian cell-based inactivated vaccines as well as insect cell-based recombinant

hemagglutinin (HA) vaccines were administered in trivalent (TIV) or quadrivalent (QIV) formats to young, healthy adults and serum samples were collected just before and 28 days after immunization, then hemagglutination inhibition assays (HAI) and surface plasmon resonance (SPR) binding assays were performed. A subset of subjects received vaccines in both consecutive seasons, and those samples were further analyzed for the influence of repeat vaccination.

The authors identify a trend toward higher HAI titers to H1N1 and H3N2 HAs in subjects receiving recombinant HA vaccine that did not reach statistical significance. Additionally, they found that binding to the HA1 domain mirrored the HAI trends. From dissociation curves, they conclude that overall serum antibody affinity increased in response to the HA1 domain of all vaccine components but did not change in response to the HA2 domain of H1N1 HA. Finally, they find that among individuals who received vaccines 2 years in a row, they observe decreased affinity maturation as measured by SPR.

Such data sets as this one, directly comparing multiple manufacturing platforms, are rare, and analysis of these sera is a crucial piece of data for the influenza vaccine development field at large. The findings reported in figure 1, comparing HAI titers of each of the groups in each of the years, are very informative and useful for the influenza field at large, as they may reflect strengths and weaknesses of each of these platforms, and suggest that recombinant HA vaccines have a chance to improve upon egg-based and mammalian cell-based inactivated vaccines.

One flaw in the manuscript is some confusion about exactly which subjects received which vaccines over the two years (and which simply reported having received a vaccine the year prior outside the study. This could be clarified by improving the labeling of Table 2, or the supporting text to more explicitly describe what is listed in each column, and to highlight exactly which of those listed in that table are included in which of the later analyses.

The authors would benefit from caution against overstating the findings of a complex analysis on a small number of samples. Their statement (in the conclusions, abstract and title of the manuscript) that the SPR dissociation data definitively show that repeat vaccination reduces affinity maturation seems hasty considering the limitations of the assay and the sample size.

SPR is a powerful technology that has the ability to reveal much about antibody/antigen interactions, however, dissociation curves are difficult to interpret based on experimental conditions employed in this analysis. Because HA/serum binding is not a 1:1 interaction (HA has up to 3 binding sites, full antibodies each have 2) drawing conclusions about affinity of antibodies found in whole serum samples is not straightforward. A more interpretable experiment might be to isolate IgG, then treat to create Fabs, and test those on SPR, giving closer to the ideal 1:1 interaction ratio. In addition, the density of antigen on the chip was high for this type of analysis—dropping to the range of 50-100 RU would also make the data more interpretable.

The idea that dissociation curves are dependent only on affinity, with no contribution of concentration, is also debatable. If I am interpreting it correctly, in supplemental figure 3, for example, the slope of the dissociation curves for each of the different dilutions of sera do appear to be different (the only change from one tracing to the next is serum concentration), and it is from those slopes that affinity is mathematically estimated.

Due to these technical issues, several of the conclusions the authors attribute to reduced affinity maturation may simply reflect reduced concentration of those antibodies produced in response to vaccination. In fig 4 for example, panels A, B and C look very similar to H, J, and K, but the reader of course cannot match up individual subjects across assays. The same could be said for fig 5. When individual subjects' HAI fold changes are compared to their off-rate fold changes, do they correlate?

The return to “baseline” off-rates by day 0 of the second year of the study is unexpected but could make sense the dissociation is actually reflecting HA1-specific antibody concentrations going back to baseline. The manuscript would benefit from a stronger focus on results and conclusions drawn from the HAI data and the SPR binding data, and less focus on the dissociation data, even though the idea that HA2 (stalk) does not induce affinity maturation/germinal centers does make sense.

Using SPR to interrogate overall serum binding to the HA0 or HA1 domains of the various proteins is informative. The HA2 domain binding results and conclusions make sense in that one would imagine individuals are already multiply-primed to the conserved epitopes of H1, H3 and B HA stalk and full-length antigens would not be so likely to induce boosting to the stalk when a head is present to dominate the immune response. Why were data for H3 and B HA2 domains not shown? It may also be useful in analyzing results on repeat vaccines to point out which strains of H3N2 and B appeared in vaccines the season prior to the opening of this study, and how much, if any, they differ from the strains used during the study.

Overall, the dataset presented here is a valuable one, and analysis of the serum samples by HAI titer and SPR binding to whole HA or individual domains is useful and compelling. The small subgroup of repeatedly vaccinated subjects is an excellent cohort, and extending to those who reported prior year vaccine is reasonable if volunteer reports are reliable. If these data, rather than the off rate/dissociation data, remained the focus of the manuscript, its conclusions would be stronger.

Reviewer #3:

Remarks to the Author:

Khurana and colleagues examined antibody responses to three commercially available influenza vaccines over two seasons and assessed the magnitude and quality of the responses to the different vaccine antigens. They used hemagglutination inhibition assays to measure the magnitude of responses and surface plasmon resonance (SPR) to assess for affinity maturation of the antibody responses (by assessing ‘off’ rates) to HA1 and an HA2 recombinant proteins. Using these methods, they found that persons who had received an inactivated influenza vaccine in the prior year were less likely to have evidence of affinity maturation of vaccine-associated, HA1-directed antibodies than those who had not received an influenza vaccine in the preceding year.

The description of the clinical study is inadequate. Additional information should be provided, including a description of eligibility criteria, the determination of past vaccination (documented or by history), the continuation of subjects from study year 1 to 2, the numbers of subjects to be enrolled (convenience sample vs based upon power calculation), and so on. It appears that the entire cohort was also divided into smaller groups for analysis (e.g., consecutive year vaccination); how these divisions were made and what the various groups analyzed need to be more clearly stated in the manuscript.

Comparisons between years 1 and 2 are made repeatedly, with ‘significant’ responses being seen more frequently in year 2. However, many more subjects were enrolled in the 2nd year in each vaccine group, making inadequate power a concern for much of the year 1 data.

Vaccine antigens in years 1 and 2 differed for the H3N2 and B strains; the figures should identify which antigens were used for the different analyses as it appears that for the SPR studies more than one strain’s rHA1 could have been used.

The authors describe antibody affinity throughout the manuscript, but they neither convincingly

demonstrate that they are measuring monovalent interactions nor do they actually measure individual antibody affinities, instead assessing average dissociation ('off') rates for polyclonal sera. While this approach does not detract from what they are proposing to demonstrate, the language used is inaccurate and should be modified.

Lines 88-93 – the paragraph has an incomplete sentence. Please modify.

Lines 105-106 – what was the randomization scheme (1:1:1) and how was the randomization schedule generated? Did study subjects receive the same type of vaccine in each study year? This information appears to be answered later in the manuscript but should be provided earlier when describing the study scheme and how subjects were divided for analysis.

Lines 111-116 – this information is redundant and was provided in an earlier paragraph. It only needs to be provided once and can be deleted from lines 85-93.

Lines 126-132 – please provide a reference for the HI methodology.

Line 132 – the presentation of titer should be done in a consistent fashion. In this line, titer is presented as a dilution (1:2) while in Figure 1 [geometric mean] titer is presented as the reciprocal of the dilution. This reviewer prefers the latter format. Please modify for consistency throughout the manuscript.

Line 152 – how was HA0 for H1N1pdm09 obtained and expressed?

Lines 182-186 – “All statistical calculations were performed using ANOVA” contradicts the earlier statement that “Differences between groups (p-values) were examined for statistical significance by the multiple comparison adjustment using Bonferroni method as well as Kruskal-Wallis non-parametric analysis.” Please clarify. There are also other statistical methods that appear to have been used in the report that are not described in the Statistical Methods section; please provide this information.

Line 183 – please confirm whether the P values in the text and figure legends represent values after Bonferroni method corrections. It appears that multiple comparisons were made within and between groups

Lines 199-202 – it appears from the results presented later that, despite randomization, there were baseline differences between the groups. Please provide basic demographic and vaccine history information for each of the study groups and for each year of the study. Were there significant differences observed between the groups?

Line 196 – change 'manufactured' to 'generated' or 'induced'

Table 1 – it is not clear why there are more subjects in Year 2 of the study than Year 1, based upon the study design described in the Methods. Please clarify how subjects were determined to be eligible and were enrolled.

Lines 209-213 and Figure 1 legend (lines 662-670) –the figure shows box plots, but the legend does not describe what the box plots represent. It is not clear what the average HI titers are from the figure – do the middle lines in the boxes represent median (traditional for box plots), arithmetic averages, or geometric means? How many persons were excluded from the analyses based upon repeat vaccination? Apparently, pair-wise comparisons were made, but this is not described in the

Statistical Methods section – what type of pair-wise test was performed? Do the numbers between D0 and D28 represent geometric mean fold rise? It is hard to believe that a 270x fold change (panel B, Fluzone) is not statistically significant, unless the numbers are so small that statistical comparison is not possible (e.g., non-parametric test used instead of a parametric test). The numbers of subjects and statistical methods used should be described in greater detail to help the reader understand the results.

Table 1 – it is not clear why subjects were stratified based upon seroprotective titers at day 0. Please provide the rationale.

Lines 227-230 – what concentrations of protein were used and how was it determined that the binding of the polyclonal antibodies would be via monovalent binding?

Lines 240-245 – only a subset of sera appear to have been analyzed in Figures 2 and 3 based upon the information shown in Table 1. For example, there are only 4 paired sera for Fluzone in year 1, but there were 9 subjects enrolled in that group, five of whom had 4-fold or higher antibody rises from day 0 to day 28. There were also paired sera in year 2 that were not analyzed. Why were some paired sera excluded and how might these exclusions have biased the data interpretation?

Lines 255-257 – “The average post vs. pre-vaccination binding titers reached statistical significance for the FluBlok groups in both year 1 and 2 and for the Fluzone group only on year 1...” What does this mean – that binding titers were different from pre- to post-vaccination?

Lines 265-268 – “However, the change in binding to all three HA1 domains following vaccination could not be directly predicted by the pre-vaccination antibody binding, suggesting a disconnect between serum antibodies and circulating B cells (memory/naïve) that can respond to the seasonal vaccine” What is the basis of this statement? What analysis was done and with which datasets?

Line 282 – measurement of affinity requires monovalent interactions; the authors do not adequately describe how this was achieved either in this manuscript or in reference #11. It is incorrect to state that the authors measured antibody affinity, especially in the context of analyzing polyclonal sera.

Line 326 – the meaning of this sentence is unclear.

Table 2 – it is unclear what the total columns represent. How are these totals different from those in Table 1?

Lines 333-337 – were the persons in the repeat vaccination group similar demographically to those in Figure 3? How are the persons in Figure 3 different? Did they not receive an influenza vaccine in the prior year – if so, this should be clearly stated in the manuscript.

Figure 4A – fold change is an incomplete representation of the data, since fold change is strongly influenced by the pre-vaccination HI titer. The authors noted that baseline serum HI titers were higher in year 2, so lower fold changes might be expected. It would be more informative to show the pre and post antibody levels in both years, much as is done for the off-rates in panels D-G.

Lines 344-346 – the findings described here contrast with the increasing antibody avidity noted by Eidem et al. (Vaccine 2015;33:4146) following repeated vaccination with the H1N1pdm09 antigen. Please comment here or in the discussion.

Lines 420-423 – the conclusions about the potential impact of FluBlok are problematic given the ~3-

fold higher HA antigen dosage in the vaccine. It has been demonstrated previously that increased HA content can increase the level and breadth of the antibody response generated (e.g., see Keitel et al. Vaccine 2008;198:1016). The potential effect of HA dosage on antibody responses must be addressed in the Discussion.

Figure 6 – since the off-rate appears to have a ceiling the fold-change off-rate will be influenced by the pre-vaccination sera's off-rate. Can the analysis of fold-change off-rate (the authors' measure of affinity maturation) be controlled for considering pre-vaccination serum in assessing the effect of prior vaccination? Does the significant impact of prior vaccination persist?

Table 1 – it is inappropriate to call persons with HI titers <40 as seronegative. Seronegative has a different meaning in the influenza literature. Please change. Also note that those persons with a titer of 40 would be considered both seropositive and seronegative based upon the definitions in footnotes a and b.

Table 1 – each cell of data appears to represent a number followed by percentage in parentheses. However, the percentage has different reference points depending on the column (total N for #seroneg and #seropos vs #seroneg and #seropos for responder columns). What is represented in each column should be more clearly defined. Percentages should also be expressed in whole numbers – there are not enough subjects studied to warrant expressing percentages to the tenth values.

Supplementary figure 1 – what were the concentrations of protein used in the SPR experiments, and how much monoclonal antibody was immobilized on the sensor chips?

Figure 2 – The figure legend indicates that pairwise comparisons were made, and asterisks placed to show level of significance. However, pairwise comparisons can only be made pre to post (within the same individual) and the horizontal bars appear to show comparisons between different vaccines. Please clarify. Also, whatever statistical method(s) was(were) used is not identified in the Statistical Methods section.

Figure 5 – $38+23=61$. What happened to the other 24 participants (of the 85)?

Figures 5 and 6 – there appear to be mean (arithmetic) lines with error bars in some of the panels. These are not described in the figure legend. It is also not appropriate to show arithmetic means (e.g., panel 6A) for geometrically distributed data.

Supplementary Figure 3 – why is a sensorgram from serum #3 shown at a dilution of 1:250 and the other two curves are for less dilute serum #2?

Response to Reviewers :

Reviewer #1:

The manuscript submitted by Khurana et al. is well written, and the logic is easy to follow. The study describes findings relating to affinity maturation in the context of serial vaccination using three seasonal influenza vaccines, based on egg produced antigens, cell line produced antigens, and a vaccine used on a recombinant product. No major difference was observed in terms of overall HI or binding titers (while the FluBlok trended a bit higher, this vaccine also has a higher dose of antigen). The authors conclude that affinity maturation could be observed for all three vaccines (and each of the three components) against the HA1, but not against HA2. Finally the authors find reduced affinity maturation in the second season studied, focusing on the repeat vaccinees. The authors conclude that they have identified an important impact of repeat vaccination on antibody-affinity maturation following vaccination, which may contribute to lower vaccine effectiveness of seasonal influenza vaccines.

While the study is well executed and described, I have several conceptual concerns about the design and interpretation of these experiments.

1. The use of such an early post-vaccination timepoint (day 28) samples to address this issue has some inherent problems. Multiple groups have studied plasma blast responses after influenza vaccination, and it stands to reason that a lot of the serum antibody observed at day 28 derives from these cells. As far as I am aware, it remains unclear if these cells have undergone a germinal center reaction, or represent short-lived, acutely induced cells. In my opinion, using a later time point for these studies would be more informative for the conclusions made in this manuscript.

Response: Based on reviewer's comment we performed additional SPR assays to measure antibody binding and antibody affinity to the HA proteins on day 180 following vaccination for year 1 study participants (no such samples available from year 2 study). The data has been added to revised figure 2 A-E, Figure 3 A-E, and fig 4 D-G.

Following descriptions have been added to Results:

Lines 342-344:

In year 1, day 180 post-vaccination samples demonstrated downward trends in antibody binding, but differences were not statistically different from D28 titers for most participants.

Lines 377-381:

Importantly, by day 180, antibody dissociation rates against the HA1 domains trended upwards (i.e. faster dissociation of antigen-antibody complexes) indicating loss/decay of the higher affinity antibodies at later time-points post-vaccination. This trend was more profound for the anti-H1 HA1- bound antibodies (Fig. 3, Panel B).

2. The lack of affinity maturation in the repeat vaccinees is only based on 16 donors, spread across the three different vaccine groups. Given the inherent variability of the data this might be a little on the low end for the interpretations made.

Response: It is true that there are 16 individuals who received the same vaccine in both years. However, 51 out of 101 participants in these vaccine studies have received an influenza vaccination in the prior year. (See revised Table 2). As is shown in Figs. 5 and 6, there is clear negative impact of prior year vaccination on antibody affinity for HA1 binding antibodies across vaccine modalities.

We also added the following statement to the Discussion (lines 618-623):

We acknowledge the limitation of the current study that was conducted as an observational descriptive study. Future prospective randomized studies with individuals with known vaccination history will be conducted to further explore the contribution of antigen platform, and antigen dose to antibody affinity maturation. It will be of interest to compare the rates of somatic hypermutations (SHM) in HA1-specific plasma cells isolated early vs. late time points after yearly vaccination.

3. Without additional repertoire analyses of vaccine induced cells, it is not clear if the observed differences derives from a lack of affinity maturation, or recruitment of a different set of B cells into the response the second year. And again, if the findings observed derive from long or short lived plasma cells. This concern is amplified by the findings shown in figure 4D, where the off rate is reduced after vaccination (affinity maturation) in season 1, but then at day 0 for season 2, seems to have come back up to a similar value as for baseline of year 1.

Response: Based on reviewer's comment we performed additional SPR assays to measure antibody affinity to the HA domains on day 180 following vaccination for year 1 study participants (no such samples available from year 2 study). The data has been added to revised Figure 4 D-G. (see response to comment 1)

We expanded the Discussion to address question raised by the reviewer (lines 586-596):

These findings suggest that the high-affinity antibodies generated after year 1 vaccination probably were derived from terminally differentiated post-GC plasma cells that were exhausted and undergone apoptosis, with minimal seeding of the bone marrow.

Short-term memory B cells that could re-seed the GC along with naïve B cells were apparently also negatively affected by repeat vaccination. Thus, large fraction of high affinity plasma cells generated after the initial vaccination may be short lived, and the B cell repertoire returns to the baseline before next year vaccination. These antibodies are of lower affinity because they are (or may be) specific for another HA based on previous exposure to influenza antigens (through infection or vaccination). The possibility of other immune mechanisms including immune exhaustion or immune competition cannot be ruled out.

4. Finally, the use of the bacterially produced antigen constructs is a concern. While the authors do show binding of a handful of monoclonal antibodies to key epitopes of the recombinant proteins, it is not clear that they truly represent the entire repertoire of epitopes on the influenza virion, possibly incorporating additional uncertainty.

Response: For SPR analysis we have also used glycosylated HA0 derived from Insect cells made by Protein Sciences that is a component of FluBlok. HA1 head of H1N1pdm09 has minimal glycosylation. In addition to MAb binding, the HA1 and HA0 proteins were tested for functional activity by binding to human RBC. The HA0 from H1N1pdm09 and all the purified rHA1 proteins from different strains agglutinated RBC to various concentrations, while the HA2 domains did not agglutinate RBC (Supp. Fig. 2). We previously demonstrated that several recombinant HA1 domains produced using bacterial system resembled native viral HA by EM, formed functional trimers/oligomers that were fully immunogenic, generated high affinity antibodies that protected ferrets from influenza challenge with pandemic strains(Ref. 13, 14,15, 17). The HA1 domains could also adsorb the majority of neutralizing antibodies from post-vaccination polyclonal antibodies in human plasma (Ref 11).

This information has been added to material and methods (lines 203-213):

To confirm that the rHA1 proteins also formed oligomers (similar to the native spike hemagglutinins on virions) hemagglutination assay was performed with human red blood cells (RBC). The HA0 from H1N1pdm09 and all the purified rHA1 proteins from different strains agglutinated RBC to various concentrations, while the HA2 domains did not agglutinate RBC (Suppl. Fig. 2). We previously demonstrated that several recombinant HA1 domains produced using bacterial system resembled native viral HA in EM, formed functional trimers/oligomers that were fully immunogenic, generated high affinity antibodies and protected ferrets from influenza challenge with pandemic strains^{13, 14, 15, 18}. The HA1 domains could also adsorb the majority of neutralizing antibodies from post-vaccination polyclonal antibodies in human plasma ¹¹.

Reviewer #2 (Remarks to the Author):

In this manuscript, “Comparison of antibody responses following different influenza vaccine platforms in humans: Repeat vaccination reduces antibody affinity maturation,” Khurana et. al. interrogate a very unique set of human sera from individuals immunized over two consecutive years with three different licensed seasonal influenza vaccines, each one produced via a different platform. Egg-based and mammalian cell-based inactivated vaccines as well as insect cell-based

recombinant hemagglutinin (HA) vaccines were administered in trivalent (TIV) or quadrivalent (QIV) formats to young, healthy adults and serum samples were collected just before and 28 days after immunization, then hemagglutination inhibition assays (HAI) and surface plasmon resonance (SPR) binding assays were performed. A subset of subjects received vaccines in both consecutive seasons, and those samples were further analyzed for the influence of repeat vaccination.

The authors identify a trend toward higher HAI titers to H1N1 and H3N2 HAs in subjects receiving recombinant HA vaccine that did not reach statistical significance. Additionally, they found that binding to the HA1 domain mirrored the HAI trends. From dissociation curves, they conclude that overall serum antibody affinity increased in response to the HA1 domain of all vaccine components but did not change in response to the HA2 domain of H1N1 HA. Finally, they find that among individuals who received vaccines 2 years in a row, they observe decreased affinity maturation as measured by SPR.

Such data sets as this one, directly comparing multiple manufacturing platforms, are rare, and analysis of these sera is a crucial piece of data for the influenza vaccine development field at large. The findings reported in figure 1, comparing HAI titers of each of the groups in each of the years, are very informative and useful for the influenza field at large, as they may reflect strengths and weaknesses of each of these platforms, and suggest that recombinant HA vaccines have a chance to improve upon egg-based and mammalian cell-based inactivated vaccines.

Response: We highly appreciate the positive comments and acknowledging the significant importance of this study.

1. One flaw in the manuscript is some confusion about exactly which subjects received which vaccines over the two years (and which simply reported having received a vaccine the year prior outside the study. This could be clarified by improving the labeling of Table 2, or the supporting text to more explicitly describe what is listed in each column, and to highlight exactly which of those listed in that table are included in which of the later analyses.

Response: In this study, subjects were asked if they had received influenza vaccine in the previous year, and randomization was stratified based on previous vaccine history. We did not attempt to verify personal vaccine history by review of medical records. In addition, subjects who had participated in the previous year were allowed to participate in the second year of the study and received the same vaccine they had received in the first year. As suggested we have revised Table 2 to add information on demographics, prior year vaccination and repeat vaccinators for both year 1 and 2.

2. The authors would benefit from caution against overstating the findings of a complex analysis on a small number of samples. Their statement (in the conclusions, abstract and title of the manuscript) that the SPR dissociation data definitively show that repeat vaccination reduces affinity maturation seems hasty considering the limitations of the assay and the sample size.

Response: It is true that there are 16 individuals who received the same vaccine in both years. However, 51 out of 101 participants in these vaccine studies have received an

influenza vaccination in the prior year. (See revised table 2 and Source data Table). As shown in Figs. 5 and 6, there is clear negative impact of prior year vaccination on antibody affinity for HA1 binding antibodies across vaccine modalities.

We also added the following statement to the Discussion (lines 618-623):

We acknowledge the limitation of the current study that was conducted as an observational descriptive study. Future prospective randomized studies with individuals with known vaccination history will be conducted to further explore the contribution of antigen platform, and antigen dose to antibody affinity maturation. It will be of interest to compare the rates of somatic hypermutations (SHM) in HA1-specific plasma cells isolated early vs. late time points after yearly vaccination.

3. SPR is a powerful technology that has the ability to reveal much about antibody/antigen interactions, however, dissociation curves are difficult to interpret based on experimental conditions employed in this analysis. Because HA/serum binding is not a 1:1 interaction (HA has up to 3 binding sites, full antibodies each have 2) drawing conclusions about affinity of antibodies found in whole serum samples is not straightforward. A more interpretable experiment might be to isolate IgG, then treat to create Fabs, and test those on SPR, giving closer to the ideal 1:1 interaction ratio. In addition, the density of antigen on the chip was high for this type of analysis—dropping to the range of 50-100 RU would also make the data more interpretable.

The idea that dissociation curves are dependent only on affinity, with no contribution of concentration, is also debatable. If I am interpreting it correctly, in supplemental figure 3, for example, the slope of the dissociation curves for each of the different dilutions of sera do appear to be different (the only change from one tracing to the next is serum concentration), and it is from those slopes that affinity is mathematically estimated.

Response: We appreciate reviewer’s attention to details and insight into the importance and relevance of SPR measured antibody affinity.

We have added the following sentence for clarification:

Introduction (lines 93-97):

For SPR, protein density on chips were optimized that assure monovalent interactions of antibodies to the surface antigens. Technically, since antibodies are bivalent, the proper term for their binding to multivalent antigens like viruses is avidity, but here we use the term affinity throughout since we measured primarily monovalent interactions.

In our earlier studies, we have optimized SPR conditions with ligand density such that binding of either MAb or mixture of MAbs and their Fab counterparts displayed similar dissociation kinetics in SPR.

Material and Methods (Lines 223-226)

Samples of freshly prepared sera at 10-, 50- and/or 250-fold dilutions were injected at a flow rate of 50 μ L/min (300-sec contact time) for association, and dissociation was performed over a 600 second interval (at a flow rate of 50 μ L/min) (Supp. Fig. 3).

Results Lines 357-363:

In the SPR system, antigen-antibody association kinetics is influenced by both antibody concentration and antibody affinity. However, the dissociation rates of antigen-antibody complexes, under conditions that favor monovalent interaction of each antibody with the HA antigen on the sensor chip, primarily reflect the inherent average affinity of the bound polyclonal antibodies¹¹. The sensorgram for a representative 10- and 50-fold dilution of human serum sample is shown in Supplementary figure 3.

4. Due to these technical issues, several of the conclusions the authors attribute to reduced affinity maturation may simply reflect reduced concentration of those antibodies produced in response to vaccination. In fig 4 for example, panels A, B and C look very similar to H, J, and K, but the reader of course cannot match up individual subjects across assays. The same could be said for fig 5. When individual subjects' HAI fold changes are compared to their off-rate fold changes, do they correlate? The return to "baseline" off-rates by day 0 of the second year of the study is unexpected but could make sense the dissociation is actually reflecting HA1-specific antibody concentrations going back to baseline. The manuscript would benefit from a stronger focus on results and conclusions drawn from the HAI data and the SPR binding data, and less focus on the dissociation data, even though the idea that HA2 (stalk) does not induce affinity maturation/germinal centers does make sense.

Response: To answer these comments, we performed additional SPR assays to measure antibody binding and antibody affinity to the HA domains on day 180 following vaccination for year 1 study participants (no such samples available from year 2 study). The data has been added to revised figure 2 A-E, Figure 3 A-E, and fig 4 D-G. All the individual data included in the manuscript is provided in the source data table.

We also performed correlation analysis between the fold change in off-rates and change in HI titers following vaccination in all participants and this has been added in the new Figure 7.

Following descriptions have been added to Results (Lines 342-344):

In study year 1, day 180 post-vaccination samples demonstrated downward trends in antibody binding, but differences were not statistically different from D28 titers for most participants.

Lines 377-381:

Importantly, by day 180, antibody dissociation rates against the HA1 domains trended upwards (i.e faster dissociation of antigen-antibody complexes) indicating loss/decay of the higher affinity antibodies at later time-points post-vaccination. This trend was more profound for the anti-H1 HA1- bound antibodies (Fig. 3, Panel B).

Results (new figure 7; Lines 496-505):

HI seroconversion rates correlate with fold change in antibody kinetics to the HA1 globular head domain

Finally, correlation was examined between the change in antibody affinity to the isolated HA domains with the functional HI activity of the polyclonal serum antibodies following vaccination of study participants. As shown in Fig. 7, a statistically significant inverse correlation was observed between the HI fold change and the fold-change in antibody off-rates of individual study participants in both year 1 and year 2 with the HA1 domains of the vaccine strains and H1N1pdm09 HA0 (Fig. 7 A, B, D, E). The fold-change in antibody off rates to the HA2 domain of H1N1pdm09 did not correlate with fold change in HI titers (Fig. 7 C).

5. *Why were data for H3 and B HA2 domains not shown? It may also be useful in analyzing results on repeat vaccines to point out which strains of H3N2 and B appeared in vaccines the season prior to the opening of this study, and how much, if any, they differ from the strains used during the study.*

Response: In the current study, we focused on the HA1 vs. HA2 of H1N1pdm09 since there were multiple reports of high levels of stem targeting antibodies against this pandemic strain after infections or vaccinations in the 2009-2010 season. We plan to generate such HA2 reagents for H3 and B for the follow up prospective studies.

6. *Overall, the dataset presented here is a valuable one, and analysis of the serum samples by HAI titer and SPR binding to whole HA or individual domains is useful and compelling. The small subgroup of repeatedly vaccinated subjects is an excellent cohort, and extending to those who reported prior year vaccine is reasonable if volunteer reports are reliable. If these data, rather than the off rate/dissociation data, remained the focus of the manuscript, its conclusions would be stronger.*

Response: We have added the following paragraph to the Discussion (Lines 557-562):

The *in vivo* correlates of protection against influenza are probably complex. However, in the ferret model we have demonstrated direct correlation between high affinity antibodies against the HA1 domain of H7N7 following vaccination and reduction in lung viral loads after challenge with either H7N7 or H7N9 highly pathogenic avian influenza viruses²⁵. Furthermore, high titers of low avidity antibodies in infected individuals were associated with severe A(H1N1)pdm09 disease³⁵.

We also added the following statement to the Discussion (lines 618-623):

We acknowledge the limitation of the current study that was conducted as an observational descriptive study. Future prospective randomized studies with individuals with known vaccination history will be conducted to further explore the contribution of antigen platform, and antigen dose to antibody affinity maturation. It will be of interest to compare the rates of somatic hypermutations (SHM) in HA1-specific plasma cells isolated early vs. late time points after yearly vaccination.

Reviewer #3 (Remarks to the Author):

Khurana and colleagues examined antibody responses to three commercially available influenza vaccines over two seasons and assessed the magnitude and quality of the responses to the different vaccine antigens. They used hemagglutination inhibition assays to measure the magnitude of responses and surface plasmon resonance (SPR) to assess for affinity maturation of the antibody responses (by assessing ‘off’ rates) to HA1 and an HA2 recombinant proteins. Using these methods, they found that persons who had received an inactivated influenza vaccine in the prior year were less likely to have evidence of affinity maturation of vaccine-associated, HA1-directed antibodies than those who had not received an influenza vaccine in the preceding year.

1. The description of the clinical study is inadequate. Additional information should be provided, including a description of eligibility criteria, the determination of past vaccination (documented or by history), the continuation of subjects from study year 1 to 2, the numbers of subjects to be enrolled (convenience sample vs based upon power calculation), and so on. It appears that the entire cohort was also divided into smaller groups for analysis (e.g., consecutive year vaccination); how these divisions were made and what the various groups analyzed need to be more clearly stated in the manuscript.

Response: We have provided additional information in Methods, Table 2 and Source Data Table, that provide extensive information about the clinical study, demographics and prior year vaccination history.

Material and Methods

Lines 101-170:

Subjects in the study were healthy adults between 18 and 49 years of age (inclusive). Good health was determined by medical history and targeted physical examination to evaluate acute or currently ongoing chronic medical diagnoses or conditions, defined as those that have been present for at least 90 days, which would affect the assessment of the safety of subjects or the immunogenicity of study vaccinations. Subjects could be taking medications if, in the opinion of the site principal investigator or appropriate sub-investigator, they posed no additional risk to subject safety or assessment of reactogenicity and immunogenicity and did not indicate a worsening of medical diagnosis or condition. Subjects had normal vital signs at the time of enrollment.

Subjects were excluded if they reported hypersensitivity to components of the study vaccine or other components of the study vaccine or latex allergy, history of severe reactions following previous immunization with licensed or unlicensed influenza virus vaccines, history of Guillain-Barre syndrome within 6 weeks of receipt of a previous influenza vaccine, if they were pregnant or breastfeeding or intended to become pregnant during the study period, if they were immunosuppressed as a result of an underlying illness or treatment with immunosuppressive or cytotoxic drugs, or use of anticancer chemotherapy or radiation therapy within the preceding 36 months or had received immunoglobulin or another blood product within the 3 months prior to enrollment in this study, if they had active neoplastic disease defined as having received treatment within the past 5 years., if they had long-term (greater than 2 weeks) use of oral or parenteral steroids, or high-dose inhaled steroids, or if they had received an inactivated vaccine within

the 2 weeks or a live vaccine within the 4 weeks prior to enrollment in this study or plans to receive another vaccine within the next 28 days.

In both year 1 and year 2 of the study, healthy adults aged 18 to 49 years were randomized to receive one of three US licensed influenza vaccines: FluBlok, a pure hemagglutinin influenza vaccine (rHA, Protein Sciences Corp) and Fluzone, a subvirion influenza vaccine made in eggs (Sanofi) or FluCelvax, a MDCK cell culture (Seqirus) derived vaccine (Table 1 and Source Data Table). This study was sponsored by National Institute of Allergy and Infectious Diseases (NIAID) under DMID Protocol Number: 15-0055. The FluBlok vaccine contain 45 µg of each HA antigen for the virus strains contained in the vaccine, while Fluzone and Flucelvax contain 15 µg of each HA antigen for the virus strains contained in the vaccine. However, the potency assays used for Flublok and Fluzone are different, and the value of “15 µg” or “45 µg” may not be directly comparable to each other.

In the 2015-2016 season, study participants received the trivalent formulation of vaccine containing A/California/7/2009 (H1N1pdm09)-like virus, A/Switzerland/9715293/2013 (H3N2)-like virus and B/Phuket/3073/2013-like virus. In the 2016–2017 season, the participants received the quadrivalent formulation of Fluzone and FluCelvax vaccine including A/California/7/2009 (H1N1pdm09)-like virus, A/Hong Kong/4801/2014 (H3N2)-like virus, B/Brisbane/60/2008-like virus and B/Phuket/3073/2013-like virus, while the FluBlok vaccine was a trivalent formulation that included A/California/7/2009 (H1N1pdm09)-like virus, A/Hong Kong/4801/2014 (H3N2)-like virus, and B/Brisbane/60/2008-like virus.

In this study, cohort of otherwise healthy adults, many of whom were employees of the University of Rochester, were asked if they had received influenza vaccine in the previous year, and randomization was stratified based on previous vaccine history. We did not attempt to verify personal vaccine history by review of medical records. In addition, subjects who had participated in the previous year were allowed to participate in the second year of the study and received the same vaccine they had received in the first year.

Subjects were randomized at the time of enrollment using an internet-based block randomization scheme. Subjects with self-reported prior vaccination history were randomized separately from those who reported that they did not receive vaccine in the previous year. In year 2 of the study, participants who had participated in year one, were re-enrolled and assigned a new study number, but received the same vaccine type that they received in year 1 of the study. The study was late getting off the ground in year 1 and was more of a pilot run for a bigger recruitment effort in year 2. See eligibility criteria above.

For SPR, based on multiple prior human studies, and assuming similar variability in change in antibody binding and antibody affinity over time, it was calculated that we will have high power to detect differences in binding/ affinity by group with 20 - 30 participants in each of the described categories (81% for n = 20 and 93% for n =30). In 2nd year (2016-17), the number of participants for each vaccine modality was 22 - 26 per vaccine arm (Table 2).

The study samples were coded, and all the antibody assays were performed blindly. After the data were generated, samples were unblinded to perform the data analysis in this study. The protocols were evaluated by CBER/NIH Research Involving Human Subjects Committee and were conducted under RIHSC exemption #03-118B.

2. Comparisons between years 1 and 2 are made repeatedly, with 'significant' responses being seen more frequently in year 2. However, many more subjects were enrolled in the 2nd year in each vaccine group, making inadequate power a concern for much of the year 1 data.

Response: The study was late getting off the ground in year 1 and was more of a pilot run for a bigger recruitment effort in year 2. Subjects with self-reported prior vaccination history were randomized separately from those who reported that they did not receive vaccine in the previous year. In year 2 of the study, participants who had participated in year one, were re-enrolled and assigned a new study number, but received the same vaccine type that they received in year 1 of the study.

For SPR, based on multiple prior human studies, and assuming similar variability in change in antibody binding and antibody affinity over time, it was calculated that we will have high power to detect differences in binding/ affinity by group with 20 - 30 participants in each of the described categories (81% for n = 20 and 93% for n =30). In 2nd year (2016-17), the number of participants for each vaccine modality was 22 - 26 per vaccine arm (Table 2).

There were 16 individuals who received the same vaccine in both years. However, 51 out of 101 participants in these vaccine studies have received an influenza vaccination in the prior year. (See revised table 2 and Source data Table). As shown in Figs. 5 and 6, there is clear negative impact of prior year vaccination on antibody affinity for HA1 binding antibodies across vaccine modalities.

This information has been added in Lines 153-166:

Subjects were randomized at the time of enrollment using an internet-based block randomization scheme. Subjects with self-reported prior vaccination history were randomized separately from those who reported that they did not receive vaccine in the previous year. In year 2 of the study, participants who had participated in year one, were re-enrolled and assigned a new study number, but received the same vaccine type that they received in year 1 of the study. The study was late getting off the ground in year 1 and was more of a pilot run for a bigger recruitment effort in year 2. See eligibility criteria above.

For SPR, based on multiple prior human studies, and assuming similar variability in change in antibody binding and antibody affinity over time, it was calculated that we will have high power to detect differences in binding/ affinity by group with 20 - 30 participants in each of the described categories (81% for n = 20 and 93% for n =30). In 2nd year (2016-17), the number of participants for each vaccine modality was 22 - 26 per vaccine arm (Table 2).

We also added the following statement to the Discussion (lines 618-623):

We acknowledge the limitation of the current study that was conducted as an observational descriptive study. Future prospective randomized studies with individuals with known vaccination history will be conducted to further explore the contribution of antigen platform, and antigen dose to antibody affinity maturation. It will be of interest to compare the rates of somatic hypermutations (SHM) in HA1-specific plasma cells isolated early vs. late time points after yearly vaccination.

3. Vaccine antigens in years 1 and 2 differed for the H3N2 and B strains; the figures should identify which antigens were used for the different analyses as it appears that for the SPR studies more than one strain's rHA1 could have been used.

Response: As suggested all figure panels are labelled with the Strain name for antigens used in SPR.

4. The authors describe antibody affinity throughout the manuscript, but they neither convincingly demonstrate that they are measuring monovalent interactions nor do they actually measure individual antibody affinities, instead assessing average dissociation ('off') rates for polyclonal sera. While this approach does not detract from what they are proposing to demonstrate, the language used is inaccurate and should be modified.

Response: We appreciate reviewer's insight into the importance and relevance of SPR measured antibody affinity.

We have added the following sentence for clarification:

Introduction (lines 93-97):

For SPR, protein density on chips were optimized that assure monovalent interactions of antibodies to the surface antigens. Technically, since antibodies are bivalent, the proper term for their binding to multivalent antigens like viruses is avidity, but here we use the term affinity throughout since we measured primarily monovalent interactions.

Results (Lines 302-310):

To that end, serially diluted sera at 10-, 50- and/or 250-fold dilutions were injected at a flow rate of 50 $\mu\text{L}/\text{min}$ (300-sec contact time) for association, and dissociation was performed over a 600 second interval (at a flow rate of 50 $\mu\text{L}/\text{min}$) (Suppl. Fig. 3). Total antibody binding was determined directly from the serum sample interaction with rHA0, rHA1 and rHA2 protein domains of the influenza virus by SPR as described before¹¹. Antibody off-rate constants, which describe the fraction of antigen-antibody complexes that decay per second, were determined directly from the serum/plasma sample interaction with rHA0, rHA1 or rHA2 in the dissociation phase only for the sensorgrams with Max RU in range of 20-150 RU for each sera using BioRad Proteon SPR machine (Suppl. Fig. 3).

Results (Lines 357-363):

In the SPR system, antigen-antibody association kinetics is influenced by both antibody concentration and antibody affinity. However, the dissociation rates of antigen-antibody complexes, under conditions that favor monovalent interaction of each antibody with the HA antigen on the sensor chip, primarily reflect the inherent average affinity of the bound polyclonal antibodies¹¹. The sensorgram for a representative 10- and 50-fold dilution of human serum sample is shown in Supplementary figure 3.

5. *Lines 88-93 – the paragraph has an incomplete sentence. Please modify.*

Response: The sentence has been modified.

Lines 105-106 – what was the randomization scheme (1:1:1) and how was the randomization schedule generated? Did study subjects receive the same type of vaccine in each study year? This information appears to be answered later in the manuscript but should be provided earlier when describing the study scheme and how subjects were divided for analysis.

Response: Subjects were randomized at the time of enrollment using an internet-based block randomization scheme. Subjects with self-reported prior vaccination history were randomized separately from those who reported that they did not receive vaccine in the previous year. In year 2 of the study, participants who had participated in year one were re-enrolled and assigned a new study number, but received the same vaccine type that they received in year 1 of the study.

We have provided additional information in Methods, Table 2 and Source Data Table, that provide extensive information about the clinical study, demographics and prior year vaccination history

Material and Methods

Lines 153-160:

Subjects were randomized at the time of enrollment using an internet-based block randomization scheme. Subjects with self-reported prior vaccination history were randomized separately from those who reported that they did not receive vaccine in the previous year. In year 2 of the study, participants who had participated in year one, were re-enrolled and assigned a new study number, but received the same vaccine type that they received in year 1 of the study. The study was late getting off the ground in year 1 and was more of a pilot run for a bigger recruitment effort in year 2.

Results:

Lines 260-266:

Subjects were randomized at the time of enrollment using an internet-based block randomization scheme. Subjects with self-reported prior vaccination history were randomized separately from those who reported that they did not receive vaccine in the previous year. In year 2 of the study, participants who had participated in year one, were re-enrolled and assigned a new study number, but received the same vaccine type that they received in year 1 of the study.

6. *Lines 111-116 – this information is redundant and was provided in an earlier paragraph. It only needs to be provided once and can be deleted from lines 85-93.*

Response: Lines 85-93 have been deleted.

7. *Lines 126-132 – please provide a reference for the HI methodology.*

Response:

Material and Methods (Lines 173-176):

HI assays were performed in microtiter format using turkey RBCs and egg-grown, A/California/07/2009 (H1N1pdm09), A/Switzerland/9715293/13 (H3N2), B/Phuket/3073/13 (Yamagata Lineage), and B/Brisbane/60/08 (Victoria Lineage) viruses¹².

Ref 12: Noble, G. R. *et al.* Measurement of hemagglutination-inhibiting antibody to influenza virus in the 1976 influenza vaccine program: methods and test reproducibility. *J Infect Dis* 136 Suppl, S429-434 (1977).

8. *Line 132 – the presentation of titer should be done in a consistent fashion. In this line, titer is presented as a dilution (1:2) while in Figure 1 [geometric mean] titer is presented as the reciprocal of the dilution. This reviewer prefers the latter format. Please modify for consistency throughout the manuscript.*

Response: The manuscript has been revised to make it consistent.

9. *Line 152 – how was HA0 for H1N1pdm09 obtained and expressed?*

Response: For SPR analysis, we used glycosylated HA0 derived from Insect cells made by Protein Sciences that is a component of FluBlok.

Material and Methods (Lines 194-195):

The glycosylated HA0 of H1N1pdm09 produced in Sf9 insect cells was purchased from Protein Sciences Inc.¹⁸

10. *Lines 182-186 – “All statistical calculations were performed using ANOVA” contradicts the earlier statement that “Differences between groups (p-values) were examined for statistical significance by the multiple comparison adjustment using Bonferroni method as well as Kruskal-Wallis non-parametric analysis.” Please clarify. There are also other statistical methods that appear to have been used in the report that are not described in the Statistical Methods section; please provide this information.*

Response: For each data analysis the information on statistical method used by the statistician is included the respective figure legends in all figures in revised manuscript.

We have also added the information on different tests used in the Methods section (Lines 238-246):

Statistical significance was performed using One-way ANOVA by the multiple comparisons test using Bonferroni method, Kruskal-Wallis non-parametric analysis by Dunn’s multiple comparisons test, or Mann-Whitney non-parametric method. In addition to p-values, the 95% confidence intervals (CIs) of the differences between study groups are also provided. A p-value less than 0.05 was considered significant. All statistical calculations were performed in GraphPad software. Spearman correlations are reported for the calculation

of correlations between off-rate fold-changes and HI titer fold-changes across all vaccine groups in year 1 and year 2.

11. Line 183 – please confirm whether the P values in the text and figure legends represent values after Bonferroni method corrections. It appears that multiple comparisons were made within and between groups.

Response: For each data analysis the information on statistical method used by the statistician is included the respective figure legends in all figures in revised manuscript.

We have also added the information on different tests used in the Methods section (Lines 238-246):

Statistical significance was performed using One-way ANOVA by the multiple comparisons test using Bonferroni method, Kruskal-Wallis non-parametric analysis by Dunn’s multiple comparisons test, or Mann-Whitney non-parametric method. In addition to p-values, the 95% confidence intervals (CIs) of the differences between study groups are also provided. A p-value less than 0.05 was considered significant. All statistical calculations were performed in GraphPad software. Spearman correlations are reported for the calculation of correlations between off-rate fold-changes and HI titer fold-changes across all vaccine groups in year 1 and year 2.

12. *Lines 199-202 – it appears from the results presented later that, despite randomization, there were baseline differences between the groups. Please provide basic demographic and vaccine history information for each of the study groups and for each year of the study. Were there significant differences observed between the groups?*

Response: As suggested, we have revised Table 2 to add information on demographics, prior year vaccination and repeat vaccinators for both year 1 and 2. Also the source table has all the information for each individual participant.

13. *Line 196 – change ‘manufactured’ to ‘generated’ or ‘induced’*

Response: Changed to ‘generated’.

14. *Table 1 – it is not clear why there are more subjects in Year 2 of the study than Year 1, based upon the study design described in the Methods. Please clarify how subjects were determined to be eligible and were enrolled.*

Response: The study was late getting off the ground in year 1 and was more of a pilot run for a bigger recruitment effort in year 2. The detailed eligibility criteria is provided in the study design in the methods section.

We have provided additional information in Methods, Table 2 and Source Data Table, that provide extensive information about the clinical study, demographics and prior year vaccination history

Material and Methods (Lines 101-170):

Clinical Study

Subjects in the study were healthy adults between 18 and 49 years of age (inclusive). Good health was determined by medical history and targeted physical examination to evaluate acute or currently ongoing chronic medical diagnoses or conditions, defined as those that have been present for at least 90 days, which would affect the assessment of the safety of subjects or the immunogenicity of study vaccinations. Subjects could be taking medications if, in the opinion of the site principal investigator or appropriate sub-investigator, they posed no additional risk to subject safety or assessment of reactogenicity and immunogenicity and did not indicate a worsening of medical diagnosis or condition. Subjects had normal vital signs at the time of enrollment.

Subjects were excluded if they reported hypersensitivity to components of the study vaccine or other components of the study vaccine or latex allergy, history of severe reactions following previous immunization with licensed or unlicensed influenza virus vaccines, history of Guillain-Barre syndrome within 6 weeks of receipt of a previous influenza vaccine, if they were pregnant or breastfeeding or intended to become pregnant during the study period, if they were immunosuppressed as a result of an underlying illness or treatment with immunosuppressive or cytotoxic drugs, or use of anticancer chemotherapy or radiation therapy within the preceding 36 months or had received immunoglobulin or another blood product within the 3 months prior to enrollment in this study, if they had active neoplastic disease defined as having received treatment within the past 5 years., if they had long-term (greater than 2 weeks) use of oral or parenteral steroids, or high-dose inhaled steroids, or if they had received an inactivated vaccine within the 2 weeks or a live vaccine within the 4 weeks prior to enrollment in this study or plans to receive another vaccine within the next 28 days.

In both year 1 and year 2 of the study, healthy adults aged 18 to 49 years were randomized to receive one of three US licensed influenza vaccines: FluBlok, a pure hemagglutinin influenza vaccine (rHA, Protein Sciences Corp) and Fluzone, a subvirion influenza vaccine made in eggs (Sanofi) or FluCelvax, a MDCK cell culture (Seqirus) derived vaccine. This study was sponsored by National Institute of Allergy and Infectious Diseases (NIAID) under DMID Protocol Number: 15-0055 (Table 2). The FluBlok vaccine contain 45 µg of each HA antigen for the virus strains contained in the vaccine, while Fluzone and Flucelvax contain 15 µg of each HA antigen for the virus strains contained in the vaccine. However, the potency assays used for Flublok and Fluzone are different, and the value of “15 µg” or “45 µg” may not be directly comparable to each other.

In the 2015-2016 season, study participants received the trivalent formulation of vaccine containing A/California/7/2009 (H1N1pdm09)-like virus, A/Switzerland/9715293/2013 (H3N2)-like virus and B/Phuket/3073/2013-like virus. In the 2016–2017 season, the participants received the quadrivalent formulation of Fluzone and FluCelvax vaccine including A/California/7/2009 (H1N1pdm09)-like virus, A/Hong Kong/4801/2014 (H3N2)-like virus, B/Brisbane/60/2008-like virus and B/Phuket/3073/2013-like virus, while the FluBlok vaccine was a trivalent formulation that included A/California/7/2009 (H1N1pdm09)-like virus, A/Hong Kong/4801/2014 (H3N2)-like virus, and B/Brisbane/60/2008-like virus.

In this study, cohort of otherwise healthy adults, many of whom were employees of the University of Rochester, were asked if they had received influenza vaccine in the previous year, and randomization was stratified based on previous vaccine history. We did not attempt to verify personal vaccine history by review of medical records. In addition, subjects who had participated in the previous year were allowed to participate in the second year of the study and received the same vaccine they had received in the first year.

Subjects were randomized at the time of enrollment using an internet-based block randomization scheme. Subjects with self-reported prior vaccination history were randomized separately from those who reported that they did not receive vaccine in the previous year. In year 2 of the study, participants who had participated in year one, were re-enrolled and assigned a new study number, but received the same vaccine type that they received in year 1 of the study. The study was late getting off the ground in year 1 and was more of a pilot run for a bigger recruitment effort in year 2. See eligibility criteria above.

For SPR, based on multiple prior human studies, and assuming similar variability in change in antibody binding and antibody affinity over time, it was calculated that we will have high power to detect differences in binding/ affinity by group with 20 - 30 participants in each of the described categories (81% for n = 20 and 93% for n =30). In 2nd year (2016-17), the number of participants for each vaccine modality was 22 - 26 per vaccine arm (Table 2).

The study samples were coded, and all the antibody assays were performed blindly. After the data were generated, samples were unblinded to perform the data analysis in this study. The protocols were evaluated by CBER/NIH Research Involving Human Subjects Committee and were conducted under RIHSC exemption #03-118B.

15. Lines 209-213 and Figure 1 legend (lines 662-670) –the figure shows box plots, but the legend does not describe what the box plots represent. It is not clear what the average HI titers are from the figure – do the middle lines in the boxes represent median (traditional for box plots), arithmetic averages, or geometric means? How many persons were excluded from the analyses based upon repeat vaccination? Apparently, pair-wise comparisons were made, but this is not described in the Statistical Methods section – what type of pair-wise test was performed? Do the numbers between D0 and D28 represent geometric mean fold rise? It is hard to believe that a 270x fold change (panel B, Fluzone) is not statistically significant, unless the numbers are so small that statistical comparison is not possible (e.g., non-parametric test used instead of a parametric test). The numbers of subjects and statistical methods used should be described in greater detail to help the reader understand the results.

Response: No samples were excluded from the analysis. Additional information has been provided for clarification.

Figure 1 legend has been modified to:

The box and whisker plot show the median value and the minimum and maximum values. The pairwise comparison of serum titers within each group that were statistically significant with p-values of <0.05 (*), or <0.005 (**), or <0.001 (***) are shown using non-parametric Krustal-Wallis with multiple comparisons analysis (Dunn's analysis) without confidence intervals. Fold change in geometric mean of HI titers between post- vs pre-

vaccination HI titers for each vaccine group are indicated for each vaccine group. Source data are provided as a Source Data file.

For each data analysis the information on statistical method used by the statistician is included the respective figure legends in all figures in revised manuscript.

We have also added the information on different tests used in the Methods section (Lines 238-246):

Statistical significance was performed using One-way ANOVA by the multiple comparisons test using Bonferroni method, Kruskal-Wallis non-parametric analysis by Dunn's multiple comparisons test, or Mann-Whitney non-parametric method. In addition to p-values, the 95% confidence intervals (CIs) of the differences between study groups are also provided. A p-value less than 0.05 was considered significant. All statistical calculations were performed in GraphPad software. Spearman correlations are reported for the calculation of correlations between off-rate fold-changes and HI titer fold-changes across all vaccine groups in year 1 and year 2.

16. Table 1 – it is not clear why subjects were stratified based upon seroprotective titers at day 0. Please provide the rationale.

Response: This information is provided for interpretation of the data in figures to define the impact of HI titers of <40 or >40 to understand the impact of vaccination.

17. Lines 227-230 – what concentrations of protein were used and how was it determined that the binding of the polyclonal antibodies would be via monovalent binding?

Response

We have established the SPR system for measurements of monovalent interactions in multiple studies (referenced in this manuscript) .

In our earlier studies, we have optimized SPR conditions with ligand density such that binding of either MAb or mixture of MAbs and their Fab counterparts displayed similar dissociation kinetics in SPR.

Importantly, we demonstrate for each HA protein coated chip that the rate of dissociation of the polyclonal antibodies is not affected by serum dilution. See Suppl. Fig. 3 samples were ran at both 1:10 and 1:50 dilutions and the off-rates were based on multiple measurements during the dissociation phase.

We have added the following sentence for clarification:

Introduction (lines 93-97):

For SPR, protein density on chips were optimized that assure monovalent interactions of antibodies to the surface antigens. Technically, since antibodies are bivalent, the proper term for their binding to multivalent antigens like viruses is avidity, but here we use the term affinity throughout since we measured primarily monovalent interactions

Results (Lines 302-310):

To that end, serially diluted sera at 10-, 50- and/or 250-fold dilutions were injected at a flow rate of 50 $\mu\text{L}/\text{min}$ (300-sec contact time) for association, and dissociation was performed over a 600 second interval (at a flow rate of 50 $\mu\text{L}/\text{min}$) (Suppl. Fig. 3). Total antibody binding was determined directly from the serum sample interaction with rHA0, rHA1 and rHA2 protein domains of the influenza virus by SPR as described before¹¹. Antibody off-rate constants, which describe the fraction of antigen-antibody complexes that decay per second, were determined directly from the serum/plasma sample interaction with rHA0, rHA1 or rHA2 in the dissociation phase only for the sensorgrams with Max RU in range of 20-150 RU for each sera using BioRad Proteon SPR machine (Suppl. Fig. 3).

18. Lines 255-257 – “The average post vs. pre-vaccination binding titers reached statistical significance for the FluBlok groups in both year 1 and 2 and for the Fluzone group only on year 1...” What does this mean – that binding titers were different from pre- to post-vaccination?

Lines 265-268 – “However, the change in binding to all three HA1 domains following vaccination could not be directly predicted by the pre-vaccination antibody binding, suggesting a disconnect between serum antibodies and circulating B cells (memory/naïve) that can respond to the seasonal vaccine” What is the basis of this statement? What analysis was done and with which datasets?

Response: We simply stated that for individual participants the pre-vaccination HI titers were not always predictive of their response to vaccination.

19. Line 282 – measurement of affinity requires monovalent interactions; the authors do not adequately describe how this was achieved either in this manuscript or in reference #11. It is incorrect to state that the authors measured antibody affinity, especially in the context of analyzing polyclonal sera.

Response: We have provided additional information on SPR measured antibody affinity.

We have added the following sentence for clarification:

Introduction (lines 93-97):

For SPR, protein density on chips were optimized that assure monovalent interactions of antibodies to the surface antigens. Technically, since antibodies are bivalent, the proper term for their binding to multivalent antigens like viruses is avidity, but here we use the term affinity throughout since we measured primarily monovalent interactions.

In our earlier studies, we have optimized SPR conditions with ligand density such that binding of either MAb or mixture of MAbs and their Fab counterparts displayed similar dissociation kinetics in SPR.

Material and Methods (Lines 223-226)

Samples of freshly prepared sera at 10-, 50- and/or 250-fold dilutions were injected at a flow rate of 50 $\mu\text{L}/\text{min}$ (300-sec contact time) for association, and dissociation was performed over a 600 second interval (at a flow rate of 50 $\mu\text{L}/\text{min}$) (Supp. Fig. 3).

Results Lines 357-363:

In the SPR system, antigen-antibody association kinetics is influenced by both antibody concentration and antibody affinity. However, the dissociation rates of antigen-antibody complexes, under conditions that favor monovalent interaction of each antibody with the HA antigen on the sensor chip, primarily reflect the inherent average affinity of the bound polyclonal antibodies¹¹. The sensorgram for a representative 10- and 50-fold dilution of human serum sample is shown in Supplementary figure 3.

20. Line 326 – the meaning of this sentence is unclear.

Response: Sentence has been clarified in lines 415-417.

In season 2, the pre-existing anti-H1N1 HI titers were comparatively higher in 50% of participants on day 0, compared with pre-existing HI titers in season 1.

21. Table 2 – it is unclear what the total columns represent. How are these totals different from those in Table 1?

Response: Table 2 has been revised to include demographics and prior year vaccination. The total numbers in Table 1 and 2 are identical.

22. Lines 333-337 – were the persons in the repeat vaccination group similar demographically to those in Figure 3? How are the persons in Figure 3 different? Did they not receive an influenza vaccine in the prior year – if so, this should be clearly stated in the manuscript.

Response: The persons were demographically similar as described in the study design and details provided in the source data table and Table 2. The prior vaccination history for these individuals is also provided in the source data table.

Figure 3 legend:

Samples from 16 individuals (Table 2) that received repeat vaccination in both year 1 and year 2 (figure 4) were not included in the figure 3 dataset.

23. Figure 4A – fold change is an incomplete representation of the data, since fold change is strongly influenced by the pre-vaccination HI titer. The authors noted that baseline serum HI titers were higher in year 2, so lower fold changes might be expected. It would be more informative to show the pre and post antibody levels in both years, much as is done for the off-rates in panels D-G.

Response: While we agree the reviewer, however, representing individual pre- and post-vaccination titers was even more confusing and hard to interpret to make conclusion. We have shown the HI data in Figure 1. We have provided the complete raw data in the source data table and summarized information in Table 1.

We also added Figure 7 in which we correlated the individual change in antibody off rates with the change in HI titers across all three vaccine platforms:

Results (new figure 7; Lines 496-505):

HI seroconversion rates correlate with fold change in antibody kinetics to the HA1 globular head domain

Finally, correlation was examined between the change in antibody affinity to the isolated HA domains with the functional HI activity of the polyclonal serum antibodies following vaccination of study participants. As shown in Fig. 7, a statistically significant inverse correlation was observed between the HI fold change and the fold-change in antibody off-rates of individual study participants in both year 1 and year 2 with the HA1 domains of the vaccine strains and H1N1pdm09 HA0 (Fig. 7 A, B, D, E). The fold-change in antibody off rates to the HA2 domain of H1N1pdm09 did not correlate with fold change in HI titers (Fig. 7 C).

24. Lines 344-346 – the findings described here contrast with the increasing antibody avidity noted by Eidem et al. (*Vaccine* 2015;33:4146) following repeated vaccination with the H1N1pdm09 antigen. Please comment here or in the discussion.

Response: We appreciate the reviewer’s comment. We address the manuscript of Eidem et al at length in our modified Discussion:

It is interesting to compare our findings with those of Eidem et al that reported persistence and avidity maturation of antibodies to A(H1N1)pdm09 in healthcare workers (HCW) following repeated annual vaccinations (2009-2011)⁴⁴. In that study, HCW were first vaccinated in 2009 with an AS03 adjuvanted H1N1pdm09 vaccine (Pandemrix; 3.75 mcg HA/dose), followed in 2010 and 2011 by unadjuvanted TIV containing 15 mcg HA/dose. While most subjects seroconverted after year 1, 2, and 3 vaccinations, the highest HI and MN GMT were measured after the first year vaccination. The apparent increase in antibody avidity between 2009 and 2011 (as measured in HA1-ELISA with or without NaSCN treatment) probably suggest that primary vaccination with the adjuvanted vaccine elicited long-term memory B cells that undergo further maturation and differentiation following year 2 and year 3 vaccinations. These findings are in agreement with our previous studies with oil-in-water adjuvanted pandemic influenza vaccines that demonstrated superior seroconversion titers, expanded epitope repertoires (more antibodies targeting protective epitopes in the HA1 domain), and significantly higher affinity maturation of antibodies in adjuvanted compared with unadjuvanted vaccine groups^{11, 45, 46}. In a long-term prime-boost study, individuals primed with MF59 adjuvanted H5N3 (clade 0) vaccine elicited rapid high titers and high affinity antibodies when boosted six years later with heterologous H5N1 (clade 1) vaccine, in contrast to individuals that were primed with unadjuvanted vaccine⁴⁷. Eidem et al also predicted that the improved antibody avidity is likely to result in improved protection *in vivo*⁴⁴.

25. Lines 420-423 – the conclusions about the potential impact of FluBlok are problematic given the ~3-fold higher HA antigen dosage in the vaccine. It has been demonstrated previously that increased HA content can increase the level and breadth of the antibody response generated (e.g., see Keitel et al. *Vaccine* 2008;198:1016). The potential effect of HA dosage on antibody responses must be addressed in the Discussion.

Response:

We have included additional information about antigen content in the methods Section:

Lines 132-136:

The FluBlok vaccine contain 45 µg of each HA antigen for the virus strains contained in the vaccine, while Fluzone and Flucelvax contain 15 µg of each HA antigen for the virus strains contained in the vaccine. However, the potency assays used for Flublok and Fluzone are different, and the value of “15 µg” or “45 µg” may not be directly comparable to each other.

26. Figure 6 – since the off-rate appears to have a ceiling the fold-change off-rate will be influenced by the pre-vaccination sera’s off-rate. Can the analysis of fold-change off-rate (the authors’ measure of affinity maturation) be controlled for considering pre-vaccination serum in assessing the effect of prior vaccination? Does the significant impact of prior vaccination persist?

Response: It is possible that high pre-vaccination off-rate can have an impact on antibody affinity maturation, especially in the case of high affinity anti-HA2 antibodies prior to vaccination.

Therefore, we show the pre-vaccination antibody affinity in Figure 3-5. However, in at least for half of repeat vaccinated individuals (Figure 5) the pre-vaccination anti-HA1 affinity at day 0 was similar for both Yr 1 and Yr 2. Yet we found less affinity maturation following second year vaccination compared with the increase in antibody affinity for after first year vaccination.

27. Table 1 – it is inappropriate to call persons with HI titers <40 as seronegative. Seronegative has a different meaning in the influenza literature. Please change. Also note that those persons with a titer of 40 would be considered both seropositive and seronegative based upon the definitions in footnotes a and b.

Response: The footnotes for Table has been revised:

**Seronegatives were defined as individuals with pre-vaccination (day 0) HI titers of ≤ 20
Seropositives were defined as individuals with pre-vaccination (day 0) HI titers of ≥ 40 .**

28. Table 1 – each cell of data appears to represent a number followed by percentage in parentheses. However, the percentage has different reference points depending on the column (total N for #seroneg and #seropos vs #seroneg and #seropos for responder columns). What is represented in each column should be more clearly defined. Percentages should also be expressed in whole numbers – there are not enough subjects studied to warrant expressing percentages to the tenth values.

Response: Table 1 has been modified according to reviewer suggestion.

29. *Supplementary figure 1 – what were the concentrations of protein used in the SPR experiments, and how much monoclonal antibody was immobilized on the sensor chips?*

Response: 1 mcg/ml Mab was captured on the chip and 1 mcg/ml of the proteins were used in SPR experiment.

30. Figure 2 – The figure legend indicates that pairwise comparisons were made, and asterisks placed to show level of significance. However, pairwise comparisons can only be made pre to post (within the same individual) and the horizontal bars appear to show comparisons between different vaccines. Please clarify. Also, whatever statistical method(s) was(were) used is not identified in the Statistical Methods section.

Response:

Legend to figure 2 have been modified to include:

The pairwise comparison of serum titers was analyzed using ordinary one-way ANOVA analysis, assuming Gaussian distribution for parametric test and was corrected for multiple comparisons using Bonferroni's method. Statistically significant with p-values of <0.05 (*), <0.005 (), or <0.001 (***) are shown. Source data are provided as a Source Data file.**

For each data analysis the information on statistical method used by the statistician is included the respective figure legends in all figures in revised manuscript.

We have clarified in Material and methods (Lines 238-246):

Statistical significance was performed using One-way ANOVA by the multiple comparisons test using Bonferroni method, Kruskal-Wallis non-parametric analysis by Dunn's multiple comparisons test, or Mann-Whitney non-parametric method. In addition to p-values, the 95% confidence intervals (CIs) of the differences between study groups are also provided. A p-value less than 0.05 was considered significant. All statistical calculations were performed in GraphPad software. Spearman correlations are reported for the calculation of correlations between off-rate fold-changes and HI titer fold-changes across all vaccine groups in year 1 and year 2.

31. *Figure 5 – 38+23=61. What happened to the other 24 participants (of the 85)?*

Response: There with 16 individuals who received vaccine in both year (total of 32 datasets) and 8 individuals (of remaining 69 participants) with unknown prior year vaccination were not included in Figure 5 dataset.

These numbers for figure 5 have been clarified:

Lines 462-464: Repeat vaccinees that received same vaccine in both years (n=16) shown in figure 4, and participants with unknown prior year vaccination history (n=8) were not included in this analysis in the figure 5.

Figure 5 legend: In the current study 38/69 individuals received inactivated seasonal influenza vaccine in the previous year (labeled inactivated), while 23/69 were not vaccinated with influenza vaccine in the previous year (labeled none).

32. Figures 5 and 6 – there appear to be mean (arithmetic) lines with error bars in some of the panels. These are not described in the figure legend. It is also not appropriate to show arithmetic means (e.g., panel 6A) for geometrically distributed data.

Response; All panel figures show geometric mean with 95% Confidence interval and this information has been added to legends of figure 5 and 6.

33. Supplementary Figure 3 – why is a sensorgram from serum #3 shown at a dilution of 1:250 and the other two curves are for less dilute serum #2?

Response: Only data used for affinity off-rate measurements is now shown to avoid confusion.

Reviewers' Comments:

Reviewer #1:

Remarks to the Author:

I am satisfied with the additional text and data provided by the authors, along with their letter.

Reviewer #2:

Remarks to the Author:

The authors have made significant improvements to the clinical portions of the manuscript, but they did not adequately address concerns about measurement of affinity of polyclonal sera by SPR dissociation analysis. The manuscript is easier to understand from a clinical perspective with the added text in the methods section. The tables (1&2) are also clearer and easier to understand, and they identify which subjects received exactly which antigens, and how many times. However, regarding the SPR analysis and this assay's ability to assess affinity of polyclonal sera, the authors have still not presented a convincing argument. Their cited reference paper (Khurana et. al. STM 2011) does not show a direct comparison of SPR dissociation analysis performed on purified IgG from polyclonal sera processed into Fabs to the same assay performed on the original polyclonal sera. This type of experiment, demonstrating the comparability of SPR dissociation of the untreated sera to SPR dissociation of processed FAbs, would be more convincing, since the processed FAbs more closely approximate the 1:1 binding ratio crucial for this type of analysis. The HAI data and the SPR binding data comparing the responses of these unique clinical cohorts is still compelling, but the conclusions stated in the title, results and discussion that are drawn from the SPR dissociation analysis are overstated. Because significant conclusions of the manuscript derive from an analysis that has not been adequately validated, the manuscript is not appropriate for publication in its current form.

Reviewer #3:

Remarks to the Author:

The authors addressed most of the concerns raised in review. A few remain and are described below:

I still have questions about some of the statistical analyses performed. One-way ANOVA allows comparisons of multiple groups, and when significant differences are observed ANOVA does not identify which groups are different. The authors need to clarify what post-hoc test was performed to identify the significant differences indicated (e.g., in Figure 2).

For the data in Figure 1, , because values are being measured at different times for the same individual and for different groups, it seems that a repeated measures nonparametric test (Friedman's test) is more appropriate than a Kruskal-Wallis test. This may explain the inability to identify 270x-fold changes (pre-post) as significantly different (Fluzone, H3N2, year 1).

For the data in Figure 2, because values are being measured at different times for the same individual and for different groups, it seems that a repeated measures ANOVA would be more appropriate.

The repeated measures analysis where ANOVA or Kruskal-Wallis was used applies to other data in the manuscript where statistical comparisons are made.

Response to Reviewers :

Editorial Comments:

Reviewers' comments:

Reviewer #1:

I am satisfied with the additional text and data provided by the authors, along with their letter.

Response: We appreciate Reviewer's acceptance of our revised manuscript

Reviewer #2:

The authors have made significant improvements to the clinical portions of the manuscript, but they did not adequately address concerns about measurement of affinity of polyclonal sera by SPR dissociation analysis. The manuscript is easier to understand from a clinical perspective with the added text in the methods section. The tables (1&2) are also clearer and easier to understand, and they identify which subjects received exactly which antigens, and how many times. However, regarding the SPR analysis and this assay's ability to assess affinity of polyclonal sera, the authors have still not presented a convincing argument. Their cited reference paper (Khurana et. al. STM 2011) does not show a direct comparison of SPR dissociation analysis performed on purified IgG from polyclonal sera processed into Fabs to the same assay performed on the original polyclonal sera. This type of experiment, demonstrating the comparability of SPR dissociation of the untreated sera to SPR dissociation of processed FAbs, would be more convincing, since the processed FAbs more closely approximate the 1:1 binding ratio crucial for this type of analysis. The HAI data and the SPR binding data comparing the responses of these unique clinical cohorts is still compelling, but the conclusions stated in the title, results and discussion that are drawn from the SPR dissociation analysis are overstated. Because significant conclusions of the manuscript derive from an analysis that has not been adequately validated, the manuscript is not appropriate for publication in its current form.

Response:

To confirm the antibody kinetics measured under optimized SPR conditions represent primarily the monovalent interactions between the antibody-antigen complex, IgG were purified from the post-vaccination serum and used to prepare Fab molecules and evaluated for binding to HA0 and HA1 in the SPR. The binding off rates of the IgG and Fabs interaction with H1N1pdm09 HA1 or HA0 were very similar in spite of the large difference in the size (molecular weight) of the bound IgG and Fab molecules. The data has been added as new Suppl. Fig. 4. The details have been provided in Methods and results sections.

Material and Methods (lines 214-218):

Purification of IgG from serum and preparation of Fab molecules

IgG was purified from serum using Protein A chromatography per manufacturer's instructions (Pierce/Thermofisher). Purified IgG was digested with Papain and the cleaved Fc was removed using Nab Protein A Plus Spin column kit (Thermofisher) and Fab fraction was collected as the flow-through fraction.

(lines 241-246):

To confirm that the intact polyclonal IgG interacts with HA via monomeric interaction under the defined SPR conditions, binding kinetics of purified IgG from post-vaccination sera and Fab fragments were compared. To that end, 10 µg/mL each of purified IgG and purified Fab fractions of serum sample were analyzed for binding to HA0 and HA1 proteins under optimized conditions in SPR as described above (Suppl. Fig. 4).

Results (lines 347-354):

Furthermore, to ascertain the antibody kinetics measured under optimized SPR conditions represent primarily the monovalent interactions between the antibody-antigen complex, IgG were purified from the post-vaccination serum and used to prepare Fab molecules and evaluated for binding to HA0 and HA1 in the SPR. The antigen-antibody binding off-rates of the IgG and Fab interaction with H1N1pdm09 HA1 or HA0 were very similar in spite of the difference in the size (molecular weight) of the bound IgG and Fab molecules. (Suppl. Fig. 4).

Reviewer #3:

The authors addressed most of the concerns raised in review. A few remain and are described below:

I still have questions about some of the statistical analyses performed. One-way ANOVA allows comparisons of multiple groups, and when significant differences are observed ANOVA does not identify which groups are different. The authors need to clarify what post-hoc test was performed to identify the significant differences indicated (e.g., in Figure 2).

For the data in Figure 1, because values are being measured at different times for the same individual and for different groups, it seems that a repeated measures nonparametric test (Friedman's test) is more appropriate than a Kruskal-Wallis test. This may explain the inability to identify 270x-fold changes (pre-post) as significantly different (Fluzone, H3N2, year 1).

For the data in Figure 2, because values are being measured at different times for the same individual and for different groups, it seems that a repeated measures ANOVA would be more appropriate.

The repeated measures analysis where ANOVA or Kruskal-Wallis was used applies to other data in the manuscript where statistical comparisons are made.

Response:

As suggested by the reviewer, we approached the director of the Division of Biostatistics at CBER, FDA, who evaluates influenza clinical trials, to perform the appropriate biostatistical analysis for our study.

We have re-evaluated the data structure (i.e., independent vaccine groups with repeated measures at baseline and post-vaccination timepoints) and the data nature of immune response measurement in view of the study objectives. A different analysis approach has been taken as a result. The statistical methods used are described in the revised manuscript.

Immunological assay and kinetic data are typically analyzed on the logarithmic scale using a parametric method, because they generally are log-normally distributed, and also theoretically should be log-normally distributed due to the dilution scheme and data analysis method of assay procedure. Logarithmic transformation generally would normalize the data distribution, and thus a parametric approach is preferred for achieving better power.

To compare the immune responses between the three vaccine arms, we analyze two endpoints (this applies to all measurements, and all assay values were log-transformed prior to analysis):

(1) Post-vaccination values: an ANCOVA model was used, with vaccine as the main effect and age, sex, and pre-vaccination value as the covariates included in the model.

(2) Fold change (Day28/Day0): an ANCOVA model was used, with vaccine as the main effect and age, sex as the covariates in the model. In addition to comparisons between vaccine arms, the estimated fold change (least squares mean) for each vaccine group was also tested for a null hypothesis of no change (ie, log fold change = 0, or fold change =1).

Bonferroni adjustment, a very conservative multiplicity adjustment method, was used to control the type I error of multiple comparisons between vaccine groups. These endpoints and analysis methods are typically used for statistical evaluation of influenza vaccine clinical studies. Adjustment for covariates are important for this dataset, because we have observed imbalance with respect to these covariates among the vaccine groups. In most of the analyses, age and sex are significant covariates. We report the adjusted means (ie, least squares means), as they provide better estimates adjusted for bias due to imbalance in these well-recognized covariates which impact the influenza vaccine immune responses. For the analysis of post-vaccination values, baseline value is the most important covariate. The effect of prior vaccination is one of the primary interest of this study. There is a separate analysis conducted to compare between subjects received prior vaccination versus none.

The detailed statistical analyses that is now included in new methods for statistical analysis and 9 new supplementary tables. These statistical analyses have been added to the manuscript as supplementary tables 1-9.

Material and Methods (lines 248-270):

Statistical analyses

For Day28 post-vaccination HI titers, resonance units (RU), and off-rate constants, an ANCOVA model was used for comparison between the vaccine groups, adjusted for gender, age, and baseline (Day0) values. For Fold changes (Day28/Day0) in HI titer, RU, and off-rate, an ANCOVA model which includes gender and age as covariates was used for

comparison between the groups. The estimated mean fold changes for each vaccine group were also tested for significance. These analyses were performed separately for each year on subjects without repeat vaccinations in both years. The year 1 results are not reliable due to very small sample sizes after removing subjects with repeat vaccination. Analyses were performed on the logarithmic scale, as normality is generally achieved by logarithmic transformation for immunogenicity data. Least square means for each vaccine group and estimated between-group differences, along with their corresponding 95% confidence intervals, were back-transformed to obtain the geometric means for each vaccine group and the geometric mean ratios between groups. Bonferroni adjustment was used to control type I error for multiple comparisons. The effect of prior vaccination on immune response was evaluated by comparing the fold changes in HI titer and off-rate between subjects received inactivated influenza vaccine in the previous year versus none for each vaccine arm, and with year 1 and year 2 combined. Spearman correlations are reported for the relationship between off-rate fold-changes and HI titer fold-changes across all vaccine groups in year 1 and year 2. Statistical analyses were performed using SAS 9.4 and GraphPad software.

Results:

The statistical analyses have been added to the manuscript as supplementary tables 1-9.

New Supplementary Tables 1-9 were mentioned in the corresponding sections throughout the manuscript and in the figure legends.

reviewers' Comments:

Reviewer #2:

Remarks to the Author:

The authors have carried out the requested additional analysis to compare whole serum off rates to those calculated using purified IgG processed into FAbs and described their methods and conclusions in the text of the manuscript. The data shown appears promising, but only one concentration is included in the graph for the purified IgG/FAbs and serum for this comparison. If the authors show a full dilution series and use that data set to calculate the off rates rather than the single concentration as currently shown in supplemental figure 4, and that off rate is comparable to that obtained from whole serum, then the analysis would be adequate to allow for the manuscript to be published with its current conclusions.

Reviewer #3:

Remarks to the Author:

The authors have satisfactorily addressed my concerns. A few additional items can be addressed for clarity.

Line 285 – change “1:40” to “40” for consistency (reciprocal of dilution is used throughout the rest of the manuscript).

Figure legends for Figures 1, 2, 3, and 5 – it appears that significant differences between groups and pairs are indicated by horizontal bars. This information should be noted in the figure legend. Also, if the intensity of the bar (e.g., fine line vs bolded line) indicates different levels of ‘significance,’ the authors should also note this in the legend.

Response to Reviewers :

Reviewer #2 (Remarks to the Author):

The authors have carried out the requested additional analysis to compare whole serum off rates to those calculated using purified IgG processed into FAbs and described their methods and conclusions in the text of the manuscript. The data shown appears promising, but only one concentration is included in the graph for the purified IgG/FAbs and serum for this comparison. If the authors show a full dilution series and use that data set to calculate the off rates rather than the single concentration as currently shown in supplemental figure 4, and that off rate is comparable to that obtained from whole serum, then the analysis would be adequate to allow for the manuscript to be published with its current conclusions.

Response:

To answer reviewer comment, antibody kinetics measured under optimized SPR conditions represent primarily the monovalent interactions between the antibody-antigen complex, IgG were purified from two post-vaccination serum samples and used to prepare Fab molecules and serial dilution of IgG and Fab were evaluated for binding to HA1 in the SPR. The binding off rates of the serial dilutions of IgG and Fabs interaction with H1N1pdm09 HA1 were very similar in spite of the large difference in the size (molecular weight) of the bound IgG and Fab molecules. The data has been added as new Suppl. Fig. 4. The details have been provided in Methods and results sections.

Material and Methods (lines 621-626):

To confirm that the intact polyclonal IgG interacts with HA via monomeric interaction under the defined SPR conditions, binding kinetics of purified IgG from post-vaccination sera and Fab fragments were compared. To that end, serial dilutions for each of purified IgG and purified Fab fractions of serum sample were analyzed for binding to HA1 proteins under optimized conditions in SPR as described above (Supplementary Fig. 4).

Results (lines 153-161):

Furthermore, to ascertain the antibody kinetics measured under optimized SPR conditions represent primarily the monovalent interactions between the antibody-antigen complex, IgG were purified from the post-vaccination serum and used to prepare Fab molecules. Serial dilution of the IgG and the corresponding Fabs from two serum samples were evaluated for binding to HA1 in the SPR. The antigen-antibody binding off-rates of the IgG and Fab interaction with H1N1pdm09 HA1 from the two serum samples were very similar in spite of the difference in the size (molecular weight) of the bound IgG and Fab molecules. (Supplementary Fig. 4).

Reviewer #3 (Remarks to the Author):

The authors have satisfactorily addressed my concerns. A few additional items can be addressed for clarity.

Line 285 – change “1:40” to “40” for consistency (reciprocal of dilution is used throughout the rest of the manuscript.

Figure legends for Figures 1, 2, 3, and 5 – it appears that significant differences between groups and pairs are indicated by horizontal bars. This information should be noted in the figure legend. Also, if the intensity of the bar (e.g., fine line vs bolded line) indicates different levels of ‘significance,’ the authors should also note this in the legend.

Response: We appreciate Reviewer’s acceptance of our revised manuscript.

For clarity we have made the following changes

The HI titer was changed from “1:40” to “40” as suggested on line 94.

Figure legends for figures 1,2,3,4 and 5 have been revised with the following sentence to represent the horizontal bars:

“Statistically significant differences between groups and pairs are indicated by horizontal bars.:

Reviewers' Comments:

Reviewer #2:

Remarks to the Author:

The authors have satisfactorily addressed my concerns.